# Enhancing Cryptographic Primitives through Dynamic Cost Function Optimization in Heuristic Search

Oleksandr Kuznetsov [1,2,*], Nikolay Poluyanenko [2], Emanuele Frontoni [1], Sergey Kandiy [2], Mikolaj Karpinski [3,4] and Ruslan Shevchuk [5,6,*]

1. Department of Political Sciences, Communication and International Relations, University of Macerata, Via Crescimbeni, 30/32, 62100 Macerata, Italy; emanuele.frontoni@unimc.it
2. Department of Information and Communication Systems Security, School of Computer Sciences, V. N. Karazin Kharkiv National University, 4 Svobody Sq., 61022 Kharkiv, Ukraine; n.poluyanenko@karazin.ua (N.P.); sergeykandy@gmail.com (S.K.)
3. Institute of Security and Computer Science, University of the National Education Commission, 30-084 Krakow, Poland; mikolaj.karpinski@up.krakow.pl
4. Department of Cyber Security, Ternopil Ivan Puluj National Technical University, 46001 Ternopil, Ukraine
5. Department of Computer Science and Automatics, University of Bielsko-Biala, 43-309 Bielsko-Biala, Poland
6. Department of Computer Science, West Ukrainian National University, 46009 Ternopil, Ukraine
* Correspondence: kuznetsov@karazin.ua (O.K.); rshevchuk@ubb.edu.pl (R.S.)

**Abstract:** The efficiency of heuristic search algorithms is a critical factor in the realm of cryptographic primitive construction, particularly in the generation of highly nonlinear bijective permutations, known as substitution boxes (S-boxes). The vast search space of 256! (256 factorial) permutations for 8-bit sequences poses a significant challenge in isolating S-boxes with optimal nonlinearity, a crucial property for enhancing the resilience of symmetric ciphers against cryptanalytic attacks. Existing approaches to this problem suffer from high computational costs and limited success rates, necessitating the development of more efficient and effective methods. This study introduces a novel approach that addresses these limitations by dynamically adjusting the cost function parameters within the hill-climbing heuristic search algorithm. By incorporating principles from dynamic programming, our methodology leverages feedback from previous iterations to adaptively refine the search trajectory, leading to a significant reduction in the number of iterations required to converge on optimal solutions. Through extensive comparative analyses with state-of-the-art techniques, we demonstrate that our approach achieves a remarkable 100% success rate in locating 8-bit bijective S-boxes with maximal nonlinearity, while requiring only 50,000 iterations on average—a substantial improvement over existing methods. The proposed dynamic parameter adaptation mechanism not only enhances the computational efficiency of the search process, but also showcases the potential for interdisciplinary collaboration between the fields of heuristic optimization and cryptography. The practical implications of our findings are significant, as the ability to efficiently generate highly nonlinear S-boxes directly contributes to the development of more secure and robust symmetric encryption systems. Furthermore, the dynamic parameter adaptation concept introduced in this study opens up new avenues for future research in the broader context of heuristic optimization and its applications across various domains.

**Keywords:** cryptographic primitives; bijective permutations; substitution boxes; nonlinearity; heuristic search; dynamic programming; cost function optimization; cryptanalysis

## 1. Introduction and Analysis of Related Works

Robust cryptographic primitives are vital for safeguarding information in the digital age. As cyber threats continue to evolve, developing advanced encryption algorithms capable of withstanding sophisticated attacks has become a critical research endeavor [1,2]. Among these primitives, substitution boxes (S-boxes) play a pivotal role in introducing

nonlinearity to symmetric ciphers, thereby enhancing their resilience against cryptanalytic techniques such as linear and differential cryptanalysis [3,4].

The construction of optimal S-boxes, which exhibit high nonlinearity and satisfy the strict avalanche criterion, has been a central focus of cryptographic research [5,6]. However, the generation of such S-boxes is a challenging combinatorial optimization problem, owing to the vast search space and complex cryptographic properties that must be satisfied [7–9]. Classical algebraic constructions, while providing S-boxes with optimal parameters, often introduce undesirable algebraic structures that can be exploited by attackers [10,11]. On the other hand, purely random generation methods, although free from such structures, have an exceedingly low probability of yielding S-boxes with the desired cryptographic strength [6,12].

To address these challenges, researchers have increasingly turned to heuristic optimization techniques for S-box generation. These methods combine the elements of randomness and guided search to efficiently explore the vast search space while avoiding the pitfalls of purely algebraic or random constructions. Simulated annealing [6,13], genetic algorithms [14–16], and hill climbing (HC) [17,18] have emerged as popular heuristic approaches in this domain. Significant progress has been made in adapting these algorithms to the specific requirements of S-box generation, such as the incorporation of novel cost functions to evaluate the cryptographic properties of candidate solutions [19–21].

Despite these advancements, generating optimal S-boxes remains a computationally intensive task. Existing heuristic methods often require a large number of iterations to converge on solutions with the desired nonlinearity [7,19,20]. Furthermore, the effectiveness of these algorithms is highly dependent on the careful tuning of their parameters and the design of the cost function [21–23]. There is a need for more efficient and adaptive heuristic techniques that can navigate the search space more effectively and locate optimal S-boxes with fewer computational resources. Table 1 presents a comparative overview of the most prominent heuristic methods, highlighting their key features, performance metrics, and limitations.

**Table 1.** Comparison of heuristic techniques for S-box generation.

| Technique | Key Features | Performance Metrics | Limitations |
|---|---|---|---|
| Simulated Annealing [6,13]—Probabilistic search | Temperature-based acceptance criteria | Nonlinearity: 102–104, Iterations: >500,000 | High computational cost, sensitive to parameter tuning |
| Genetic Algorithms [14–16]—Population-based search | Crossover and mutation operators | Nonlinearity: 100–104, Iterations: >200,000 | Premature convergence, difficulty in maintaining diversity |
| Hill Climbing [17,18]—Gradient-based search | Local neighborhood exploration | Nonlinearity: 102–104, Iterations: >100,000 | Prone to getting stuck in local optima, dependent on initial solution |
| Particle Swarm Optimization [24] | Swarm intelligence, Velocity and position updates | Nonlinearity: 94–100 | Susceptibility to local optima |
| Our Proposed Method | Dynamic parameter adaptation, Efficient hill climbing with WCFS cost function | Nonlinearity: 104, Iterations: <50,000 | Potential for further improvement |

As evident from Table 1, existing heuristic techniques for S-box generation suffer from various limitations, such as high computational costs, sensitivity to parameter tuning, premature convergence, and susceptibility to local optima. Moreover, these methods often

require a large number of iterations to converge on solutions with the desired nonlinearity, leading to significant computational overhead.

In contrast, our proposed method introduces a novel dynamic parameter adaptation mechanism that enhances the efficiency and effectiveness of the hill-climbing algorithm. By dynamically adjusting the cost function parameters based on the search progress, our approach is able to locate optimal S-boxes with fewer iterations and higher nonlinearity compared to state-of-the-art techniques.

We validate our method through extensive experiments and comparative analyses with state-of-the-art heuristic techniques for S-box generation. The results demonstrate that our dynamic parameter adaptation strategy consistently outperforms existing approaches, achieving a 100% success rate in finding S-boxes with nonlinearity of 104 or higher while requiring significantly fewer iterations. Furthermore, we provide insights into the behavior of our algorithm and discuss the implications of our findings for the design of efficient heuristic methods in cryptography.

The comparative analysis presented in Table 1 highlights the novelty and advantages of our proposed method over state-of-the-art techniques. By addressing the limitations of existing approaches and introducing a dynamic parameter adaptation mechanism, our work contributes to the advancement of heuristic optimization techniques for S-box generation in cryptographic applications. The main contributions of this paper are as follows:

1.  We introduce a novel dynamic parameter adaptation mechanism for heuristic search algorithms in the context of S-box generation.
2.  We demonstrate the effectiveness of our approach through rigorous experiments and comparative analyses with state-of-the-art techniques.
3.  We provide insights into the behavior of our algorithm and discuss the implications of our findings for the design of efficient heuristic methods in cryptography.

The remainder of this paper is organized as follows. The background section lays the necessary theoretical foundation, covering the mathematical properties of S-boxes, heuristic search algorithms, and the Walsh–Hadamard transform. The proposed methodology introduces a dynamic parameter adaptation mechanism and an enhanced hill-climbing algorithm for efficient S-box generation. The article then presents the experimental setup, benchmark instances, performance metrics, and a detailed analysis of the results obtained from three series of experiments. The discussion section interprets the findings, compares the proposed method with state-of-the-art techniques, and addresses the limitations and future research directions. Finally, the conclusion summarizes the main contributions, emphasizes the novelty and effectiveness of the approach, and offers concluding remarks on the potential impact of the research in the field of cryptography.

## 2. Nomenclature

This section provides a comprehensive list of the symbols, notations, and abbreviations used throughout the article. Each entry includes a clear and concise definition to facilitate the reader's understanding of the presented concepts and formulas.

S—An S-box (substitution box), a basic component in symmetric key cryptography
n—The size of the S-box input in bits
m—The size of the S-box output in bits
x—An input value to the S-box
y—An output value from the S-box
$\{0;1\}^n$—The set of all possible n-bit binary strings
WHT(a,b)—Walsh–Hadamard Transform of the S-box at points a and b
a·x—The inner product of a and x in the Boolean algebra context
b·S(x)—The inner product of b and S(x) in the Boolean algebra context
N(S)—The nonlinearity of the S-box S
$\Delta$(S)—The differential uniformity of the S-box S
$\vec{H}(S)$—The histogram of the Walsh–Hadamard coefficients for the S-box S

$i_{max}$—The maximum absolute value of the Walsh–Hadamard coefficients for the S-box
$l_i$—The number of Walsh–Hadamard coefficients with an absolute value i
$S_s$—The search space in the context of heuristic search
f—The cost function or objective function in the context of heuristic search
R—A parameter of the WCFS cost function
X—A parameter of the WCFS cost function
WCFS(S)—The cost function used in the hill-climbing algorithm for S-box generation

## 3. Background

The heuristic generation of S-boxes stands as a cornerstone in modern symmetric ciphers, ensuring nonlinearity and reducing vulnerabilities to attacks [5,6,25]. This section delves into the essential terminologies and cryptographic measures of S-boxes, and explicates the heuristic methods in targeting an optimal state. Additionally, the prominence of the hill-climbing method in combinatorial optimization is elaborated, setting the stage for our subsequent discourse on the prevailing gaps in research and our innovative contribution to this domain.

### 3.1. S-Boxes (Substitution-Boxes)

In the realm of symmetric cryptography, the S-box (Substitution box) stands as a pivotal nonlinear transformation component [26,27]. Its primary role is to map input bits to output bits, introducing a level of confusion and complexity to the encryption process [28].

From a conceptual standpoint, an S-box can be visualized as a black box that takes a fixed number of bits as inputs and produces a fixed number of bits as outputs. The transformation is defined by a lookup table, which specifies the output for each possible input.

In the context of Boolean algebra, an S-box can be represented as a set of Boolean functions [29,30]. Given an m-bit input and an n-bit output, an S-box is defined by n Boolean functions, each taking m variables.

For an S-box with a size of n×m, it can be represented as [30]:

$$S:\{0;1\}^m \longrightarrow \{0;1\}^n$$

where $\{0;1\}^n$ is the domain representing all possible n-bit binary strings and $\{0;1\}^m$ is the codomain representing all possible m-bit binary strings. Specifically, we focus on S-boxes that take an 8-bit input and produce an 8-bit output, ensuring a one-to-one mapping between the input and output spaces.

To ensure an S-box is resilient against cryptanalysis, several measures have been postulated [30]:

- Bijectiveness ensures that every input has a unique output and vice versa. For an S-box to be bijective, it must be both injective (one-to-one) and surjective (onto). Mathematically, an S-box $S$ is bijective if, and only if, for every pair of distinct inputs $x$ and $y$, their outputs are also distinct, and every possible output has a corresponding input:

$$\forall x, y \in \{0,1\}^8, S(x) = S(y) \Rightarrow x = y.$$

- Nonlinearity measures the deviation of an S-box from affine functions. It provides resistance against linear cryptanalysis. The nonlinearity of an S-box is computed using the Walsh–Hadamard transform. Given an S-box $S$, its nonlinearity $N(S)$ is defined as:

$$N(s) = \frac{1}{2}(2^n - max_{a,b \neq 0}|WHT(a,b)|),$$

where $WHT(a,b)$ is the Walsh–Hadamard transform of S and is defined as:

$$WHT(a,b) = \sum_{x \in \{0,1\}^n} (-1)^{b \cdot S(x) + a \cdot x} \tag{1}$$

Here, $a \cdot x$ and $b \cdot S(x)$ are the inner products in the Boolean algebra context.

- Differential Uniformity is a measure of the resistance of the S-box against differential cryptanalysis. The lower the differential uniformity, the better the S-box is in terms of resisting differential attacks. For an S-box $S$, its differential uniformity $\Delta(S)$ is defined as:

$$\Delta(S) = max_{\Delta x, \Delta y} \left| \left\{ x \in \{0,1\}^8 : S(x) \oplus S(x \oplus \Delta x) = \Delta y \right\} \right|,$$

where $\oplus$ denotes the bitwise XOR operation.

- There are several other properties of S-boxes that are crucial for their cryptographic strength, such as the Avalanche effect, Strict Avalanche Criterion, Bit Independence Criterion, and more. Each of these properties ensures that the S-box provides good diffusion and resistance against various cryptanalytic attacks.

### 3.2. Heuristic Techniques

Heuristic search, grounded in the realm of optimization and problem solving, emerges as a pivotal strategy when faced with extensive search spaces or computationally rigorous problems [31,32]. This approach delicately toes the line between finding optimal solutions and ensuring computational viability.

Central to the heuristic search is the concept of a search space, the entire domain comprising potential solutions [33,34]. Given the complexity of certain problems, the vastness of this space can render exhaustive searches nonviable. In such landscapes, a heuristic, a rule-of-thumb technique, provides a promising avenue. It may not always guarantee the most optimal solution, but it navigates the vast space to produce an approximation in a time frame that is algorithmically acceptable.

A quintessential component that drives heuristics is the cost function or the objective function [35]. Mathematically, if we let $S_s$ be our search space, the cost function $f$ can be represented as $f\colon S_s \longrightarrow S$, mapping each candidate solution to a real value, indicative of its quality. This function not only quantifies the desirability or fitness of a solution, but also becomes the guiding light, directing the heuristic search towards regions of the search space that show potential. Furthermore, it offers a comparative landscape, allowing one solution to be weighed against another, and often serves as a termination beacon, indicating when the search should conclude.

Numerous heuristic search strategies have found prominence in the literature and practice. Among them, Genetic Algorithms draw inspiration from natural selection [33,36], encoding solutions as 'genes' and iteratively refining them over successive generations using processes akin to mutation and crossover. Another fascinating method, Simulated Annealing, borrows its philosophy from metallurgy, particularly the annealing process [36,37]. This probabilistic technique oscillates between accepting solutions of varying quality based on a dynamically adjusted temperature parameter. Yet another strategy, the Tabu Search, incorporates memory in its operations, maintaining a list of recently traversed solutions to avoid cyclical traps [35,36].

However, the hill-climbing algorithm deserves special mention [36]. Its conceptual simplicity combined with iterative refinement makes it a darling in the world of combinatorial optimization [38]. The algorithm initiates with a randomly chosen solution and steadily refines it by evaluating its neighbors in Algorithm 1. The quality of these neighbors is adjudged by the cost function, and the best among them is chosen for the next iteration. This process perseveres until no further refinement yields a better solution. Its inherent advantage lies in its straightforwardness, adaptability, and efficiency.

The hill-climbing algorithm is an iterative optimization algorithm that starts with an arbitrary solution to a problem and iteratively refines the current solution by making small changes to it. The idea is to always move towards a better solution in the search space [36,38].

---

**Algorithm 1:** Hill-Climbing Algorithm (Generalized Version)

---

Algorithm HillClimbing(search_space)
    current_solution ← initialize_random_solution(search_space)
    current_value ← evaluate(current_solution)

    while True do
        neighbors ← generate_neighbors(current_solution)

        if neighbors is empty then
            return current_solution
        end if

        next_solution ← best_solution(neighbors)
        next_value ← evaluate(next_solution)

        if next_value $\leq$ current_value then
            return current_solution
        end if

        current_solution ← next_solution
        current_value ← next_value
    end while
end Algorithm

---

Description and Comments:

1.  Initialization: The algorithm begins by initializing a solution randomly. This serves as our current reference point.
    current_solution←initialize_random_solution(search_space)
    Here, the search_space represents the entire space of possible solutions.
    The function initialize_random_solution simply picks a point (or solution) from this space at random.
2.  Evaluation: We assess the quality or fitness of our current solution using an evaluation function.
    current_value←evaluate(current_solution)
3.  Iterative Refinement:
    - The main loop of the algorithm focuses on refining the current solution.
    - In every iteration, we look at the neighboring solutions of our current state, which is performed by the generate_neighbors function. These neighbors are typically small variations of the current solution.
    - If there are no neighbors (an edge case), we simply return the current solution as our best found.
    - From these neighbors, we choose the best one (having the highest or lowest value, depending on whether it is a maximization or minimization problem). The function best_solution facilitates this selection.
    - We then evaluate this new best solution from the neighbors.
    - A critical decision point ensues. If the new solution's value is not better than our current solution, the algorithm terminates, and our current solution is returned as the best found. If the new solution is better, we shift our current solution to this new point and continue the process.
4.  Termination:
    - The algorithm terminates either when no better neighboring solutions can be found or when there are no neighbors left for the current solution.

The generation of cryptographic S-boxes, which act as vital components in symmetric key algorithms, poses a formidable challenge. The need to ensure bijectiveness and simul-

taneously achieve high nonlinearity demands robust optimization techniques. Enter the specialized hill-climbing algorithm in Algorithm 2—crafted with a discerning focus on this particular challenge [39–41]. The pseudo-code for this variant is detailed in Appendix C of the work [17].

---

**Algorithm 2:** Hill-Climbing Algorithm for S-boxes Generation (version from [17])

---

Algorithm SpecializedHillClimbing (S,NoE)
    while (NoE > 0) do
        S'←S
        Select random distinct positions i,j
        Swap outputs of S' corresponding to i and j
        if N(S') > N(S) or N(S') = N(S) and CS' < CS then
            S←S'
        end if

        NoE← NoE−1
    end while
    return S
end Algorithm

---

Description and Comments:

1. Initialization: The algorithm accepts an initial random substitution *S* and a predetermined number of solution evaluations, *NoE*, which is typically set to $10^6$, as suggested in [17].

2. Iterative Refinement:

   - During each iteration, a copy *S'* of the current solution *S* is made. Two random distinct positions, *i* and *j*, are selected from this substitution.
   - A pivotal step involves swapping the outputs of *S'* corresponding to the positions *i* and *j*. This is the essence of creating a neighboring solution in the context of S-box generation.
   - The newly formed neighbor, *S'*, is then evaluated on two primary metrics: *N(S)* (nonlinearity) and *CS'* (cost function value, as evaluated by the WCFS). If *S'* offers superior nonlinearity or, in the event of identical nonlinearity, a lower cost, it supplants the current solution *S*.

3. Termination: The algorithm halts after a preset number of evaluations, *NoE*, ensuring bounded computation.

The merit of this tailored hill-climbing approach, when synergized with the new cost function (called WCFS), is underscored by its empirical success. Remarkably, in just an average of 70,000 iterations, it was able to generate an 8 × 8 bijective S-box with a nonlinearity of 104 [17]. Further honing this technique, the same researchers managed to slash this to an even more impressive 65,933 iterations [20], edging ever closer to cryptographic perfection.

The concepts and techniques discussed in this background section lay the foundation for our proposed methodology. By understanding the mathematical properties of S-boxes, such as bijectiveness, nonlinearity, and differential uniformity, we can formulate effective cost functions and heuristic search strategies for generating optimal S-boxes. The Walsh–Hadamard transform, in particular, plays a crucial role in evaluating the nonlinearity of S-boxes and guiding the search process. Moreover, the hill-climbing algorithm, with its iterative refinement approach and specialized adaptations for S-box generation, serves as the backbone of our proposed method. Building upon these fundamental concepts and leveraging the strengths of heuristic optimization, our work aims to develop a more efficient and effective approach for constructing cryptographically strong S-boxes. In the following sections, we will delve into the details of our methodology, presenting the dynamic parameter adaptation mechanism and the enhanced hill-climbing algorithm that form the core of our contribution to the field of S-box generation.

## 4. Methodology

S-box generation has been a central pillar in the realm of cryptographic research. Our investigative endeavors are predicated upon the foundational works detailed in [17,20], where seminal results in this domain were elucidated. These prior research contributions offer an optimal search algorithm and an associated cost function, both of which lay the groundwork for our own explorations. With this canvas, we extend the paradigm by introducing refined mechanisms to optimize the search process, integrating dynamic parameter adjustments, and, consequently, ushering in a more nuanced understanding of nonlinear substitution generation.

### 4.1. Cost Function

Grounded in the research drawn from [17,20], we employ the WCFS function as our cost function, which is represented as:

$$WCFS(S) = \sum_{\substack{b \in \{0,1\}^n, b \neq 0 \\ |WHT(b,i)| > X}} \sum_{i \in \{0,1\}^n} \left(\frac{|WHT(b,i)| - X}{4}\right)^R, \tag{2}$$

where *WHT indicates the WalshHadamard transform* (see Equation (1)) and both *R* and *X* are specified real numbers.

The efficacy of any search algorithm hinges on its capacity to find the target S-box within a minimal number of iterations. Specifically, for our approach, the metric of interest is a bijective S-box that aligns with our desired nonlinearity *N(S)*. Given that evaluating the cost function is computationally intensive, our algorithm's runtime primarily corresponds to the iteration count, making each invocation of the WCFS function crucial.

### 4.2. Hill Climbing Algorithm

While leveraging the robust hill-climbing algorithm propounded in [17,20], we instituted a series of modifications aimed at refining its efficiency and precision. Central to our enhancements is the integration of novel termination conditions:

Achievement of a maximum number of inner cycles, termed as max_frozen_loops, where no discernible improvements in state—either in terms of nonlinearity augmentation or, given consistent nonlinearity, a reduction in the cost function—are realized.

Attainment of the stipulated nonlinearity, defined as target_nonlinerity.

These nuanced stop criteria sculpt our specialized version of the S-boxes search algorithm in Algorithm 3.

In any algorithmic investigation, particularly in cryptographic research, the choice of parameters is neither arbitrary nor capricious; it emerges as a decisive factor that heavily influences the efficiency and effectiveness of the underlying algorithm. Such choices are honed through empirical testing, analytical insights, and often, prior foundational work. Our decision to configure the parameters as mentioned is deeply intertwined with our previous scholarly endeavors, elucidated in [23,42,43]. Below, we undertake a meticulous discourse on our parameter selection.

Nonlinearity remains a quintessential measure of an S-box's ability to resist linear cryptanalysis. The higher the nonlinearity, the more resilient the S-box is against potential analytical attacks. Our choice to set the target nonlinearity at either 106 or 104 stems from both empirical studies and the evolving requirements of modern cryptographic applications. As examined in [43], the ramifications of varying nonlinearity levels on security were explored in-depth. Choosing 106 or 104 as target benchmarks encapsulates a balance, aiming for high resistance while also ensuring computational feasibility.

---

**Algorithm 3:** Hill-Climbing Algorithm for S-boxes Generation (Our version)

---

Algorithm EnhancedHillClimbing(S, max_loops, max_frozen_loops, target_nonlinerity)
    while (max_loops > 0) AND (n < max_frozen_loops) AND (N(S) < target_nonlinerity)) do
    S′←S
    n←0
        Select at random two different positions i and j
        Swap outputs of S′ corresponding to i and j
        if N(S′) > N(S) OR (N(S′) = N(S) AND WCFS(S′) ≤ WCFS(S)) then
            S←S′
            n←0
        else:
            n←n + 1
            max_loops← max_loops−1
    return S
end Algorithm

---

### 4.3. Parameter Selection

Heuristic algorithms, by nature, involve a degree of exploration and are inherently probabilistic. Thus, setting a bound on the number of loops is crucial to ensure that the algorithm does not perpetuate indefinitely, especially when the search space is vast and the optimum might be elusive. Our decision to set a ceiling at 2,500,000 is rooted in findings from [23,42], where the influence of iteration bounds on the convergence and performance of heuristic algorithms was explored. Too few iterations might result in premature convergence, while an excessive count risks computational wastage. Through rigorous testing, 2,500,000 emerged as an empirically robust choice, allowing the algorithm ample opportunity to navigate the search space without becoming prohibitively time consuming.

A nuanced innovation in our version of the hill-climbing algorithm is the introduction of a secondary loop threshold, max_frozen_loops. This parameter is crucial in mitigating the risks of stagnation. If the algorithm fails to improve after a sequence of iterations, it indicates potential entrapment in a local minimum. As detailed in [23,42], setting a bound on such 'frozen' iterations provides an elegant escape mechanism, ensuring that the search is not paralyzed by local optima. The 2500 count has been refined through extensive simulations, striking a balance between allowing the algorithm room to 'breathe' and ensuring timely extrication from potential stagnation points.

So, we use the hill-climbing algorithm with the following parameters:

- target_nonlinerity = 106 or 104;
- max_loops = 2,500,000;
- max_frozen_loops = 2500.

At each iteration of the algorithm, we use the change of two randomly selected positions in the S-box as mutations.

### 4.4. Hierarchical Description and Component Interaction

The proposed methodology for efficient S-box generation can be viewed as a hierarchical system composed of interacting components. At the highest level, the system comprises three main components (Figure 1): the cost function, the hill-climbing algorithm, and the parameter selection mechanism. These components work in concert to achieve the desired computational efficiency and generate high-quality S-boxes.

The cost function component, represented by the WCFS function (2), serves as the guiding force behind the search process. It evaluates the quality of candidate S-boxes by measuring their nonlinearity and other cryptographic properties. The cost function is invoked repeatedly by the hill-climbing algorithm to assess the fitness of each S-box configuration explored during the search.

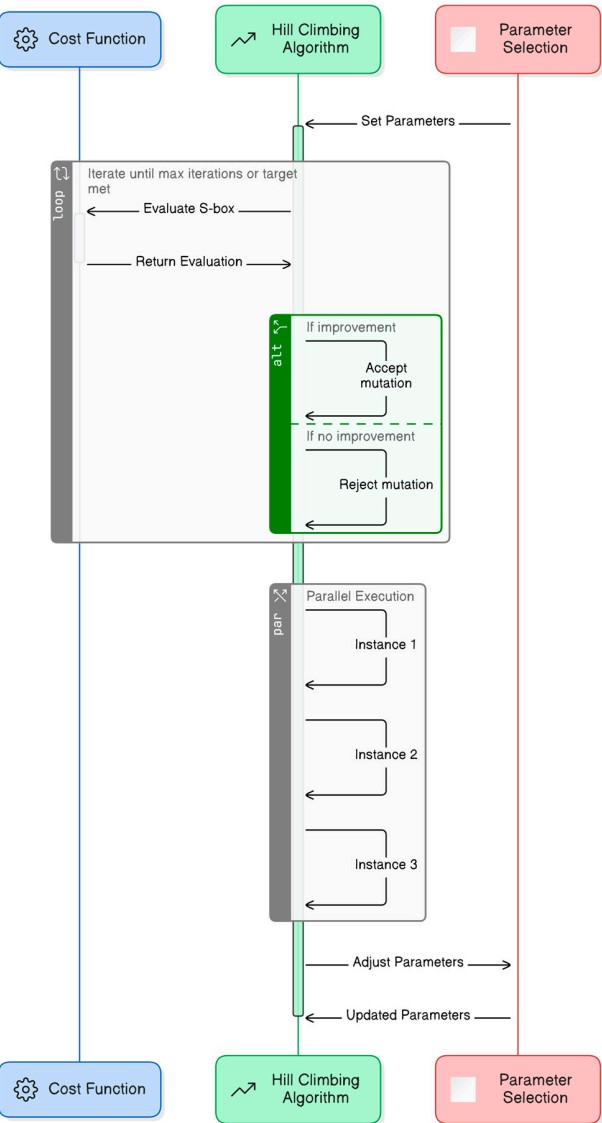

**Figure 1.** Hierarchical interaction model for optimized S-box generation.

The hill-climbing algorithm component forms the core of the methodology. It is responsible for navigating the vast search space of possible S-box configurations and iteratively refining the candidate solutions. The algorithm starts with an initial S-box and performs a series of mutations, swapping randomly selected positions to generate new configurations. The cost function is used to evaluate the quality of each new configuration, and the algorithm decides whether to accept or reject the mutation based on the improvement in the cost function value.

The parameter selection component plays a crucial role in determining the behavior and efficiency of the hill-climbing algorithm. It encompasses the choice of target nonlinearity (target_nonlinerity), the maximum number of iterations (max_loops), and the secondary loop threshold (max_frozen_loops). These parameters are carefully chosen based on empirical studies and theoretical considerations to strike a balance between exploration and exploitation in the search process.

The interaction between these components is characterized by a tight feedback loop. The hill-climbing algorithm relies on the cost function to guide its search, while the cost function's effectiveness is determined by the quality of the S-boxes generated by the

algorithm. The parameter selection component influences the behavior of the hill-climbing algorithm, which, in turn, affects the efficiency and effectiveness of the overall methodology.

To achieve computational efficiency, the proposed approach leverages the inherent parallelism of the hill-climbing algorithm. The algorithm can be easily parallelized by running multiple instances of the search process concurrently, each starting from a different initial S-box configuration. This parallel exploration of the search space allows for a more efficient utilization of computational resources and can significantly reduce the overall runtime of the methodology.

Furthermore, the dynamic adjustment of the cost function parameters within the hill-climbing algorithm contributes to the improved computational efficiency. By adapting the parameters based on the progress of the search, the methodology can effectively balance the exploration of new regions in the search space with the exploitation of promising solutions. This dynamic adaptation helps the algorithm to converge faster towards high-quality S-boxes, reducing the number of iterations required to achieve the desired nonlinearity.

Figure 1 elucidates the architecture of an optimized S-box generation methodology, highlighting a sophisticated interaction among its primary components: the cost function, the hill-climbing algorithm, and the parameter selection mechanism. This model emphasizes a dynamic and recursive evaluation process that strategically employs hill-climbing techniques and parameter adjustments to enhance cryptographic robustness and computational efficiency.

In summary, the proposed methodology achieves enhanced computational efficiency through the careful orchestration of its components. The cost function guides the search process, while the hill-climbing algorithm efficiently explores the search space. The parameter selection mechanism ensures that the algorithm strikes an optimal balance between exploration and exploitation. The inherent parallelism of the hill-climbing algorithm is exploited to accelerate the search process, and the dynamic adjustment of cost function parameters contributes to faster convergence. Through this hierarchical design and the synergistic interaction of its components, the proposed methodology is able to efficiently generate high-quality S-boxes for cryptographic applications.

## 5. Results

In this study, we conduct a series of experiments to evaluate the performance and behavior of our proposed heuristic search algorithm for generating highly nonlinear S-boxes. The experiments are designed to assess the impact of various parameter settings and cost function configurations on the algorithm's ability to find optimal S-boxes efficiently.

### 5.1. Organization of Overall Experiments

Our experimental approach is based on the analysis of the Walsh–Hadamard spectrum of the S-boxes. Let $\vec{H}(S)$ be the histogram of the Walsh–Hadamard coefficients for the S-box $S$ (as introduced in [19]). This is a vector whose value at position $i$ corresponds to the number of coefficients in position $|i|$ of the Walsh–Hadamard spectrum of the S-box $S$. Let $i_{max}$ denote the maximum (last) position in this vector with a non-zero value. Then, the maximum absolute value of the Walsh–Hadamard coefficients for the S-box $S$ is $i_{max}$, and this coefficient determines the nonlinearity of the S-box. We denote $l_i$ by the number of values of the Walsh–Hadamard coefficients for each $i$.

Reducing the value of the WCFS function narrows the histogram of the Walsh–Hadamard coefficients. We present the obtained estimates of the maximum narrowing that brings us closer to nonlinearity in $N(S) = 106$.

To assess the progress more accurately, we conducted two tests with 1000 runs of the search algorithms for the bijective S-boxes with $N(S) = 104$ each (the search algorithm parametrizations are similar to the above). In the first test, the parameters of the cost function WCFS were ($R = 11$, $X = 4$), and in the second—($R = 12$, $X = 0$). The arithmetic value $k_{itr} = 53{,}508$ was set to ($R = 11$, $X = 4$) and $k_{itr} = 53{,}025$ for ($R = 12$, $X = 0$).

The average absolute (i.e., taken for $|i|$) number of the first three maximum values of the Walsh–Hadamard coefficients was:

- for ($R = 11$, $X = 4$): $l_{|i=40|} = 607$; $l_{|i=44|} = 256$; $l_{|i=48|} = 73$;
- for ($R = 12$, $X = 0$): $l_{|i=40|} = 606$; $l_{|i=44|} = 255$; $l_{|i=48|} = 73$.

Given that the number of possible changes in two positions in the S-box is $\frac{256 \cdot 257}{2} = 32,896$, we slightly modified the search algorithm. Initially, the algorithm was executed as described above, and starting from some state (for example, if $N(S) = 102$ was found), a full search of all possible mutations (change of two positions) was performed. If an improvement in the cost function was found, it was implemented, and the full search was started again. At the same time, the criteria for stopping the search algorithm were also slightly changed:

- finding a bijective S-box with nonlinearity of 106 (never performed for this modification of the search algorithm);
- achievement of the maximum number of iterations;
- performing all possible mutations for the current S-box and not finding any improvement (almost always performed).

Thanks to the full search, we firstly guaranteed that a local minimum was found (no mutation could lead to an improvement in the cost function), and secondly, we reduced the total number of iterations by excluding random changes that were repeatedly made and did not lead to improvements.

We conducted 100 independent runs of the hill-climbing search algorithm with the above modification and the following WCFS function parameters:

- $R = 11$, $X = -16$; $-16$; $-12$; $-8$; $-4$; 0; 4; 8; 12; 16; 20;
- $R = 12$, $X = -20$; $-16$; $-12$; $-8$; $-4$; 0; 4; 8.

The histograms of the distribution of the averaged absolute values of the Walsh–Hadamard coefficients for these tests are shown in Figures 2 and 3, respectively. The results are presented on a logarithmic scale. For comparison, the last column shows similar values of the results presented in Figure 4.

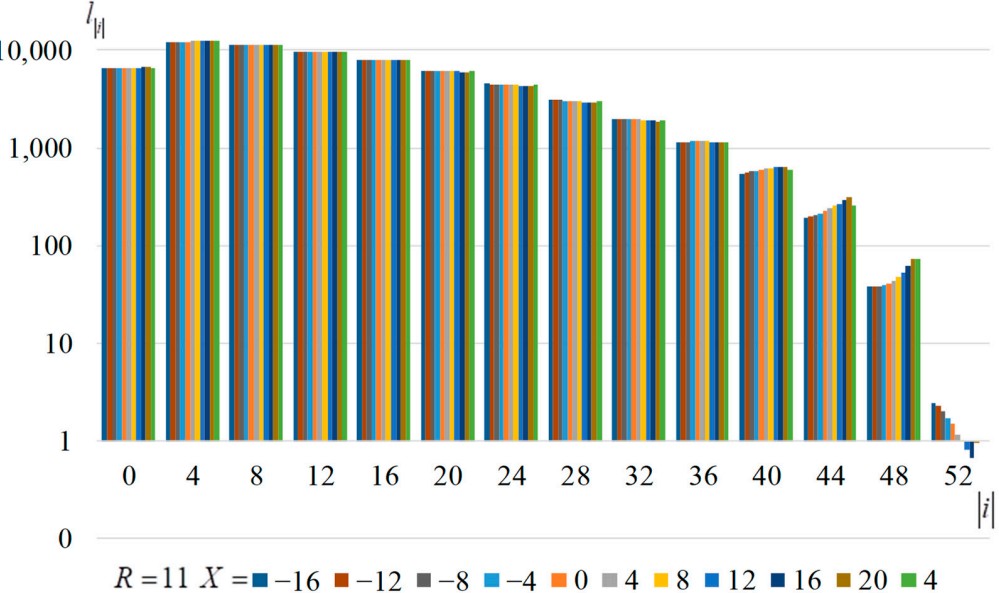

**Figure 2.** Histograms of the distribution of the averaged absolute values of the Walsh–Hadamard coefficients, when local minima are reached, for $R = 11$.

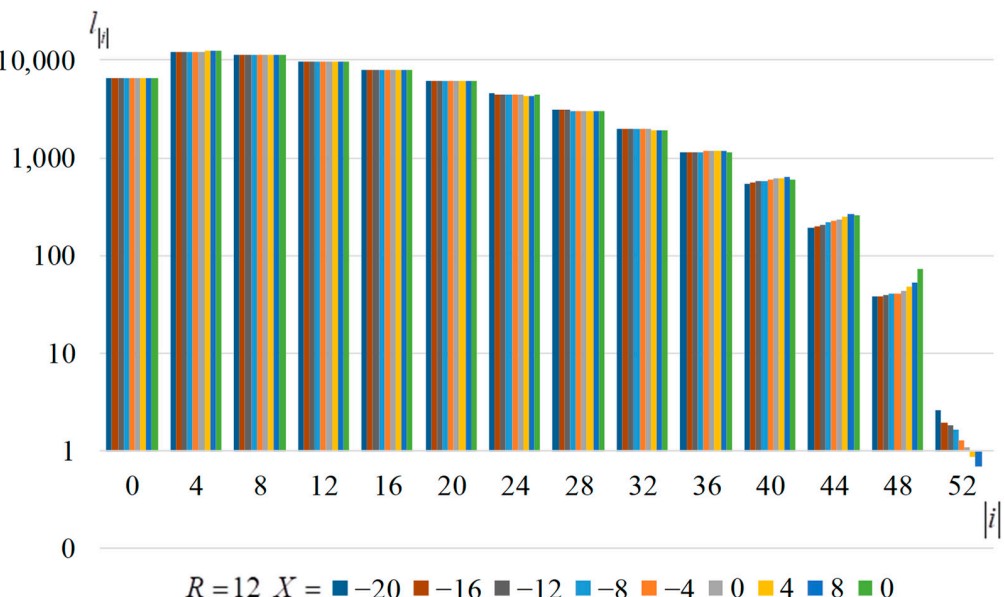

**Figure 3.** Histograms of the distribution of the averaged absolute values of the Walsh–Hadamard coefficients, when local minima are reached, for *R* = 12.

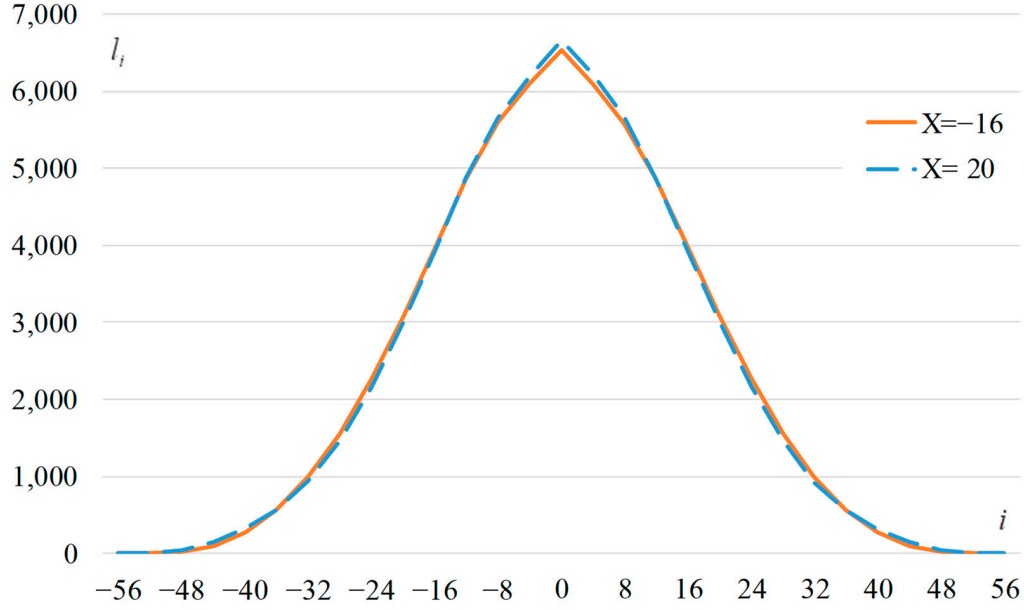

**Figure 4.** The average distribution of the Walsh–Hadamard spectral coefficients at the minimum values of the WCFS function with the parameters R = 11 and X = −16 (solid line), X = 20 (dashed line).

*5.2. Benchmark Instances and Cost Function Parameters*

In this subsection, we present the benchmark instances and cost function parameters used in our experiments. We focus on the generation of 8-bit S-boxes with high nonlinearity, specifically targeting a nonlinearity of N(S) = 104 or higher. The choice of these benchmark instances is motivated by their relevance to modern cryptographic applications and the challenges they pose for heuristic search algorithms.

Our experimental analysis reveals several key findings regarding the behavior of the search algorithm and the impact of cost function parameters on its performance:

1. With an increase in the parameter *X*, at a fixed value *R*, the variance in the graph of the distribution of the averaged absolute values of the Walsh–Hadamard coefficients increases. This leads to a narrowing of the distribution in the middle part, but a

widening in the extreme parts. For example, Figure 4 shows the distribution for two values of *X*. If we try to reduce the number $l_{|i=48|}$ by decreasing the parameter *X*, then $i_{max}$ increases, with which the nonlinearity decreases to 100 and below. If we manage to find an S-box with *N(S)* = 104, it usually leads to an increase in $l_{|i=48|}$ (see the last columns of Figures 2 and 3).

2.  To achieve a local minimum using the specified search method, an average of $k_{itr}$ = 250,000 ÷ 350,000 iterations is required, which is 5–6 times more than that to achieve only *N(S)* = 104.

3.  Considering the regions of minimal values of the average number of iterations (where the parameters correspond to the ratio *X* = 52 − 4·*R*), when the local minimum of the WCFS function is reached, the average value of the number of coefficients for the extreme positions in the Walsh–Hadamard spectrum distribution is:

-   $l_{|i=44|}$ = 190 ÷ 314;
-   $l_{|i=48|}$ = 38 ÷ 74;
-   $l_{|i=52|}$ = 2.4 ÷ 0.7.

It should be noted that when $l_{|i=48|} \approx 38$, the number of coefficients of the spectrum of *i* = 52 is $l_{|i=52|} \approx 2 \div 3$ (corresponding to *N(S)* = 102), and 1–2% have coefficients of *i* = 56 (*N(S)* = 100). When we approach *N(S)* = 104, the average number of coefficients of the spectrum of *i* = 48 is about $l_{|i=48|}$ = 60 ÷ 75. This indicates the practical impossibility of forming S-boxes of *N(S)* = 106. The goal of achieving the local minimum of $l_{|i=48|}$ = 0, using the described algorithm, is almost unattainable.

Given the above, it is necessary to further modify the search algorithm, introducing additional changes that would help to achieve the goal. We see the following ways:

-   Extending the mutation of the current S-box. Instead of changing two positions in the S-box, use three or more positions. However, if the number of possible changes in two positions in the S-box is $\frac{256 \cdot 257}{2}$ = 32,896 iterations, then using the formula for the number of placements from *n* to *k*, for the case of a three-change mutation, the possible number of mutations will be:

$$\frac{n!}{(n-k)!} = \frac{256!}{(256-3)!} = 256 \cdot 255 \cdot 254 = 16,581,120,$$

which significantly increases the complexity and time of the search. We conducted selective testing of this method and, in our opinion, it is not a rational way. Time costs increase by more than two orders of magnitude, and improvements are rarely or not at all found. In the case of a significant mutation (a change of three or more positions), the distribution of the spectrum of the Walsh–Hadamard coefficients obtained by the S-box almost always deteriorates significantly.

-   Using dynamic changes in the cost function. If we are at a local minimum for one function, this does not mean that this state of the system will be a local minimum for another function. However, it is impossible to change the cost function drastically, because it will greatly worsen the spectral distribution of the Walsh–Hadamard coefficients. We propose a slight change in the parameters of the WCFS function we have already chosen. This will be discussed in more detail below.

-   An alternative to the above point is to use several different cost functions that are somewhat similar to each other (which will not allow for significant changes in the best solution) but have different value tracks (which will lead to different values of local minima and allow for exit from them).

-   Applying the acceptance of deterioration of the cost function in the search algorithm. It is possible to modify the search algorithm using the methods of the annealing simulation algorithm, which will help to escape from the local minimum.

-   At the same time, using a combination of the above methods.

The hill-climbing algorithm was used in the following modification. Initially, the hill-climbing algorithm was performed in its "classical" form. After the first time, the

nonlinearity of *N(S)* = 102 or the number of iterations without an improvement in the cost function exceeded 32,000, and instead of randomly selecting a pair of S-box positions for mutation, a complete search of all possible mutations of the two positions was performed. If an improvement in the cost function was found, it was accepted, and the search was started again with the new current S-box. If no improvement was found, the value *X* was changed, and the search continued.

It was found that if we changed the parameters *X* and *R* simultaneously, moving along the ratio $X = 48 - 4 \times R$, then there was almost no improvement in the distribution of the spectrum of the Walsh–Hadamard coefficients. Most likely, no improvements were observed due to the fact that the shape of the spectrum distribution, while maintaining $X = 48 - 4 \times R$, did not change.

### 5.3. Experimental Environment

To ensure a rigorous and fair evaluation of our proposed algorithm, we conducted a series of experiments designed to assess its performance and behavior under various settings and in comparison with state-of-the-art techniques. The primary motivation behind our experimental design was to comprehensively investigate the effectiveness and efficiency of the dynamic parameter adaptation mechanism in the context of S-box generation.

All experiments were performed on a standard desktop computer equipped with an Intel Core i7 processor and 16 GB of RAM, providing a consistent hardware environment across all runs. The algorithms were implemented in C++ and compiled using GCC version 9.2.0.

We employed a diverse set of problem instances, including randomly generated S-boxes, to evaluate the algorithm's performance across a wide range of scenarios. The dynamic adaptation rate and initial cost function parameters were varied for our algorithm, while the recommended default settings were used for the compared techniques to maintain a fair comparison.

### 5.4. Performance Metrics

To evaluate the performance of the proposed algorithm and compare it with existing heuristic techniques, we considered three key metrics:

1.  Success Rate: The success rate represents the percentage of runs in which the algorithm successfully finds an S-box with the target nonlinearity. In our experiments, we focused on generating $8 \times 8$ S-boxes with a nonlinearity of 104. A higher success rate indicates a more reliable and effective algorithm.
2.  Average Iterations: The average number of iterations required by the algorithm to find the target S-box is a crucial metric for assessing the computational efficiency. A lower average iteration count signifies a faster convergence and more efficient search process.
3.  Runtime: The runtime metric measures the average wall-clock time (in seconds) taken by the algorithm to complete a single run. This metric provides a practical assessment of the algorithm's efficiency and scalability, considering both the computational complexity and the actual execution time.

By considering these performance metrics, we aim to provide a comprehensive evaluation of our proposed algorithm and demonstrate its effectiveness in comparison with state-of-the-art techniques. The success rate and average iterations metrics shed light on the algorithm's ability to find optimal S-boxes consistently and efficiently, while the runtime metric offers insights into its practical applicability and computational overhead.

In the following subsections, we present the experimental results obtained for each series of experiments, along with a detailed analysis and discussion of the findings. Through this rigorous experimental evaluation, we seek to validate the superiority of our dynamic parameter adaptation approach and establish its potential for generating high-quality S-boxes in cryptographic applications.

### 5.5. Visualization and Explanation of Results

5.5.1. Dynamic Parameter Adaptation: Impact on Convergence and Nonlinearity

In this series of experiments, we investigate the impact of dynamic parameter adaptation on the convergence behavior and nonlinearity of the generated S-boxes. By visualizing and analyzing the trajectories of the Walsh–Hadamard coefficient distributions across different parameter settings, we gain insights into the effectiveness of our proposed approach. In all runs of the first series of the search algorithm, the value of the parameter $R$ was fixed. The tests were performed with R = 8; 9; 10; 11; 12. The results were averaged over at least 100 runs. In each run, four cycles of changing parameter $X$ were performed in the following chain (see Table 2).

**Table 2.** Selected parameters $X$ and $R$ in the first series of experiments (the first part).

| $R$ | $X$ | | | | | | | | | | | | | | | | | | | | |
|---|---|---|---|---|---|---|---|---|---|---|---|---|---|---|---|---|---|---|---|---|---|
| 8 | 28 | 24 | 20 | 16 | 12 | 8 | 4 | 0 | 4 | 8 | 12 | 16 | 20 | 24 | 28 | | | | | | |
| 9 | 24 | 20 | 16 | 12 | 8 | 4 | 0 | −4 | 0 | 4 | 8 | 12 | 16 | 20 | 24 | | | | | | |
| 10 | 20 | 16 | 12 | 8 | 4 | 0 | −4 | −8 | −12 | −16 | −12 | −8 | −4 | 0 | 4 | 8 | 12 | 16 | 20 | | |
| 16 | 20 | 16 | 12 | 8 | 4 | 0 | −4 | −8 | −12 | −16 | −20 | −16 | −12 | −8 | −4 | 0 | 4 | 8 | 12 | 16 | 20 |
| 12 | 16 | 12 | 8 | 4 | 0 | −4 | −8 | −12 | −16 | −20 | −24 | −20 | −16 | −12 | −8 | −4 | 0 | 4 | 8 | 12 | 16 |

Figures 5–16 show the trajectory of change in the average number of absolute values of the Walsh–Hadamard spectral coefficients at the end of the algorithm for each value of $X$, at: $i = 52(a)$; $i = 48(b)$; $i = 44(c)$. They also show the average total number of iterations performed by the search algorithm since its launch ($d$).

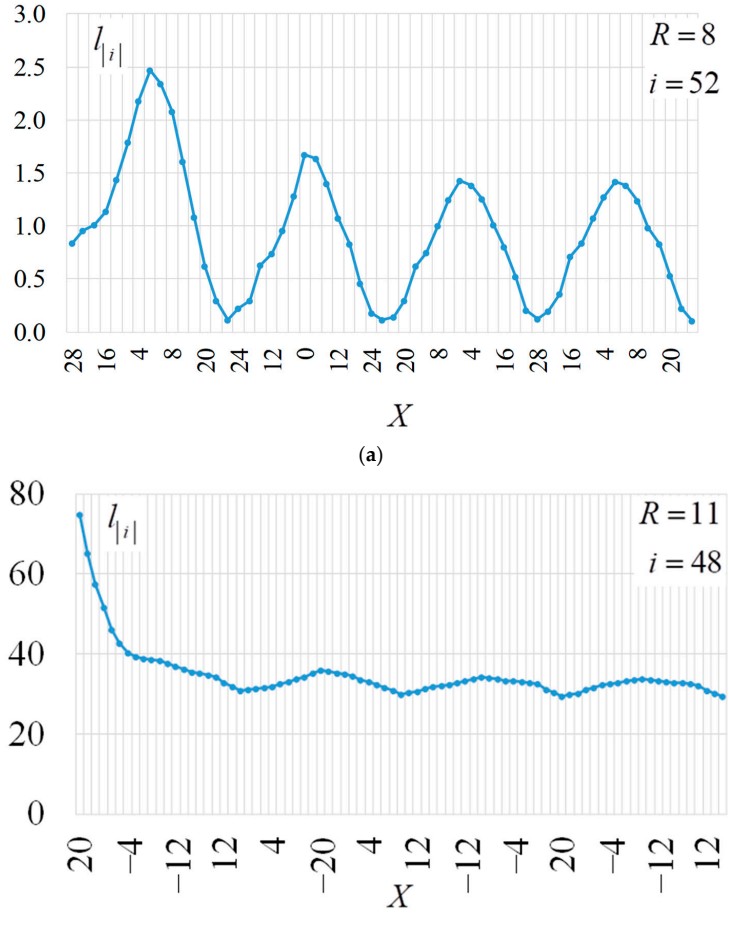

(a)

(b)

**Figure 5.** *Cont.*

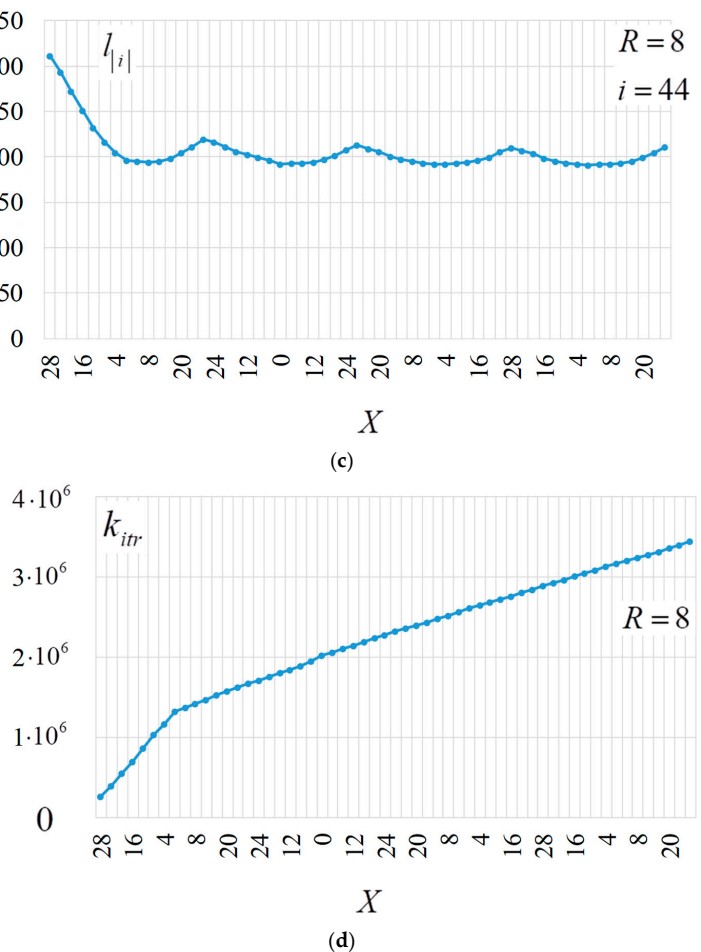

(c)

(d)

**Figure 5.** Trajectory of the $l_{|i|}$ S-box during dynamic change $X$ (4 cycles) and $R = 8$.

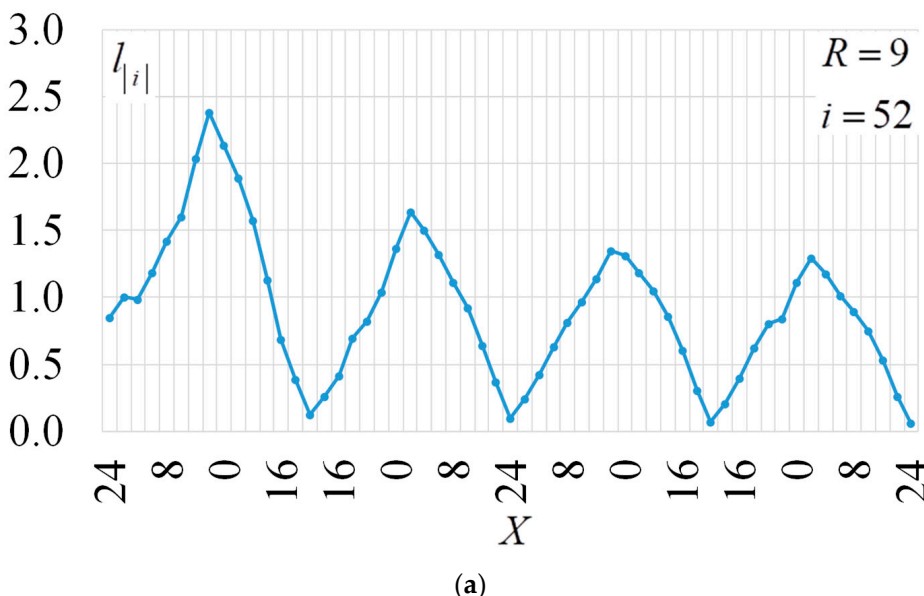

(a)

**Figure 6.** *Cont.*

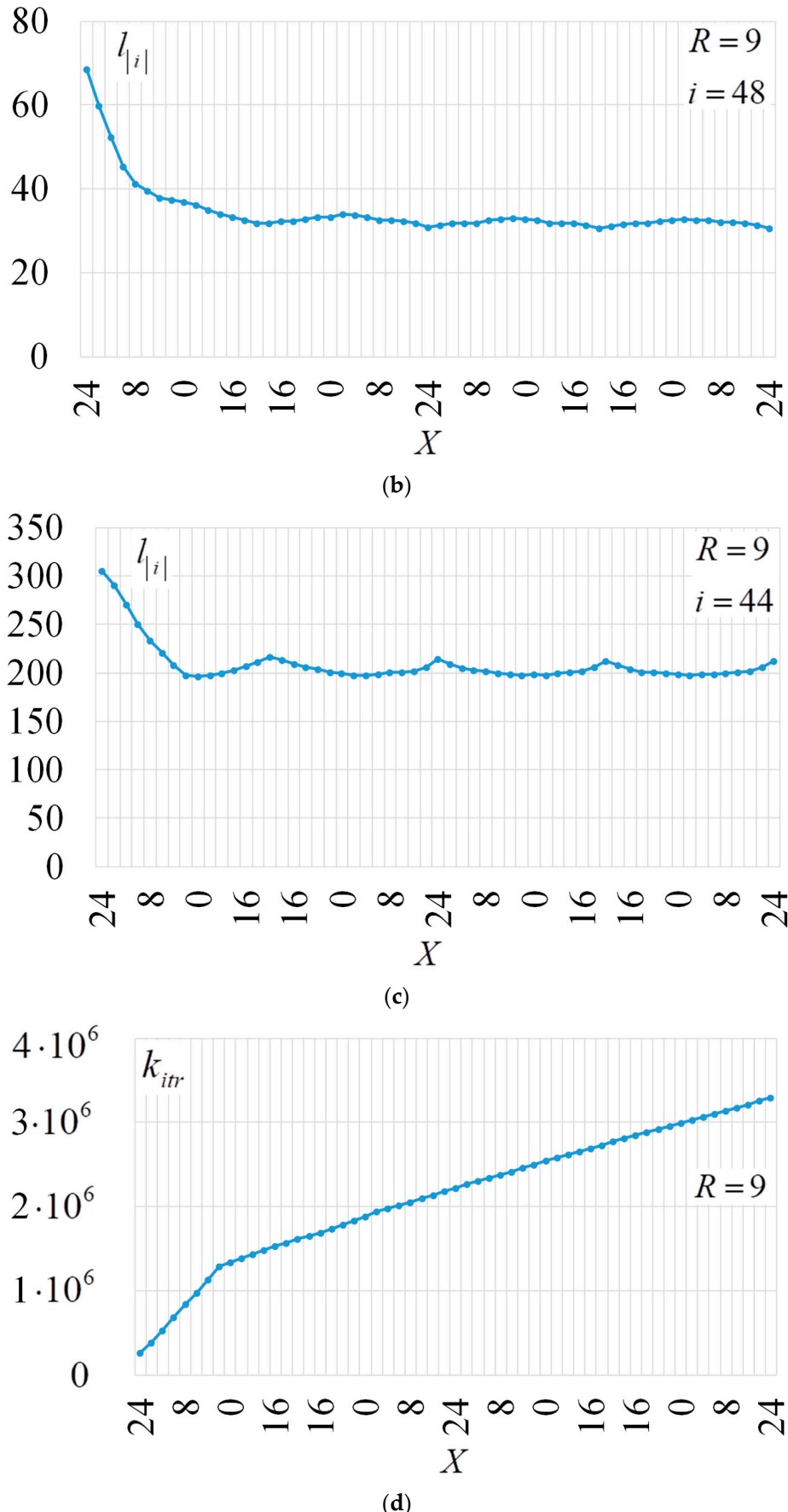

**Figure 6.** Trajectory of the $l_{|i|}$ S-box during dynamic change $X$ (4 cycles) and $R = 9$.

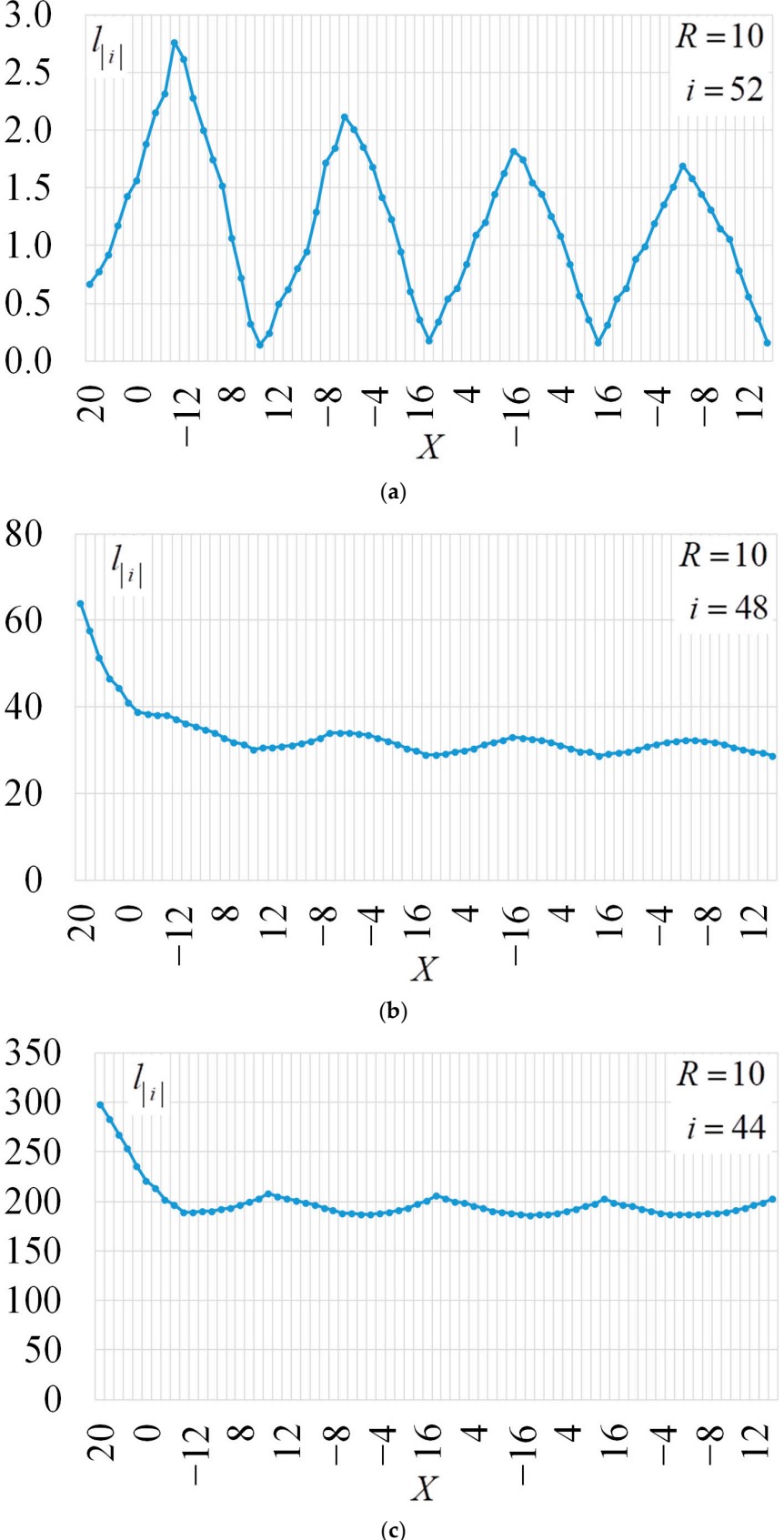

**Figure 7.** *Cont.*

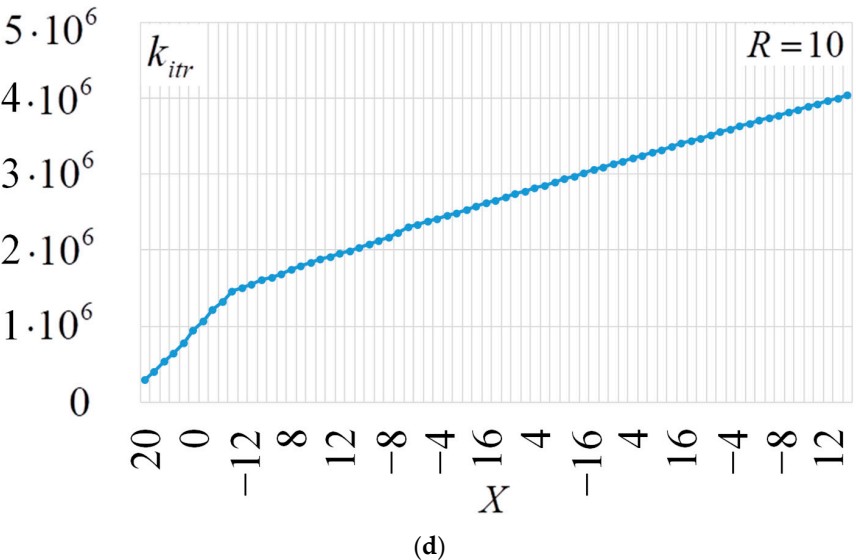

(**d**)

**Figure 7.** The trajectory of the $l_{|i|}$ S-box at dynamical change $X$ (4 cycles) and $R = 10$.

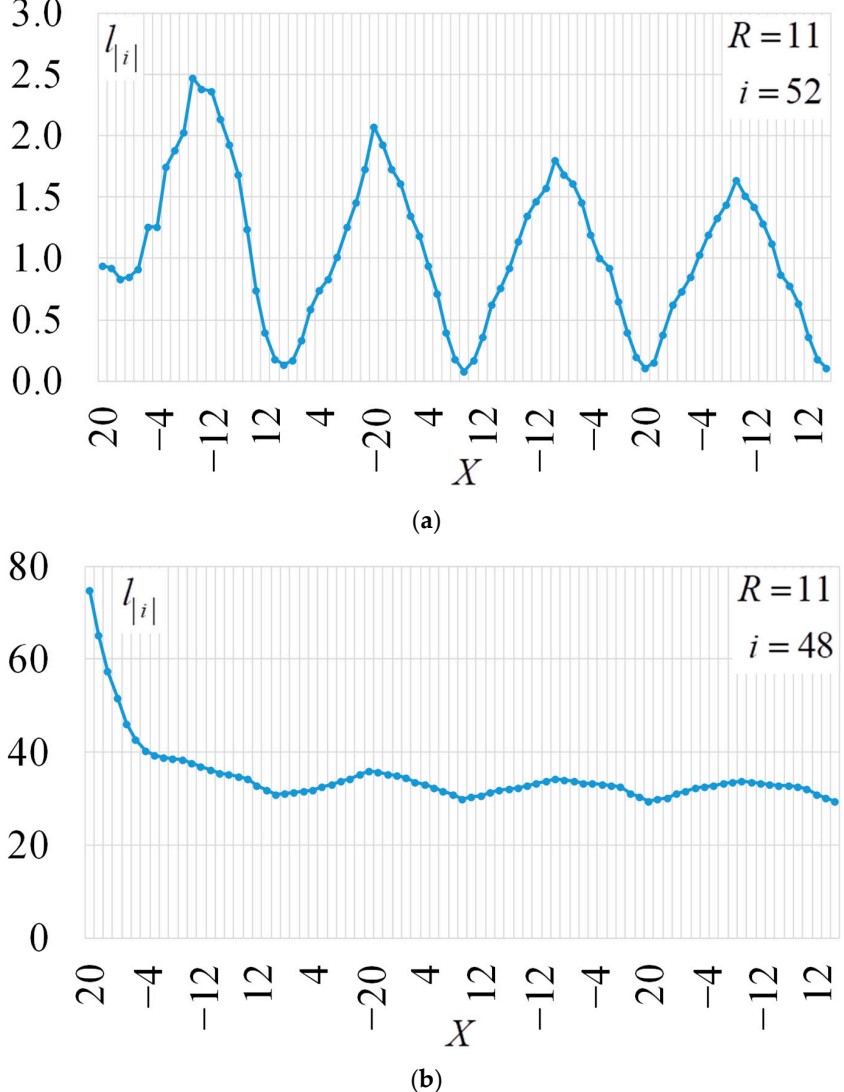

(**a**)

(**b**)

**Figure 8.** *Cont.*

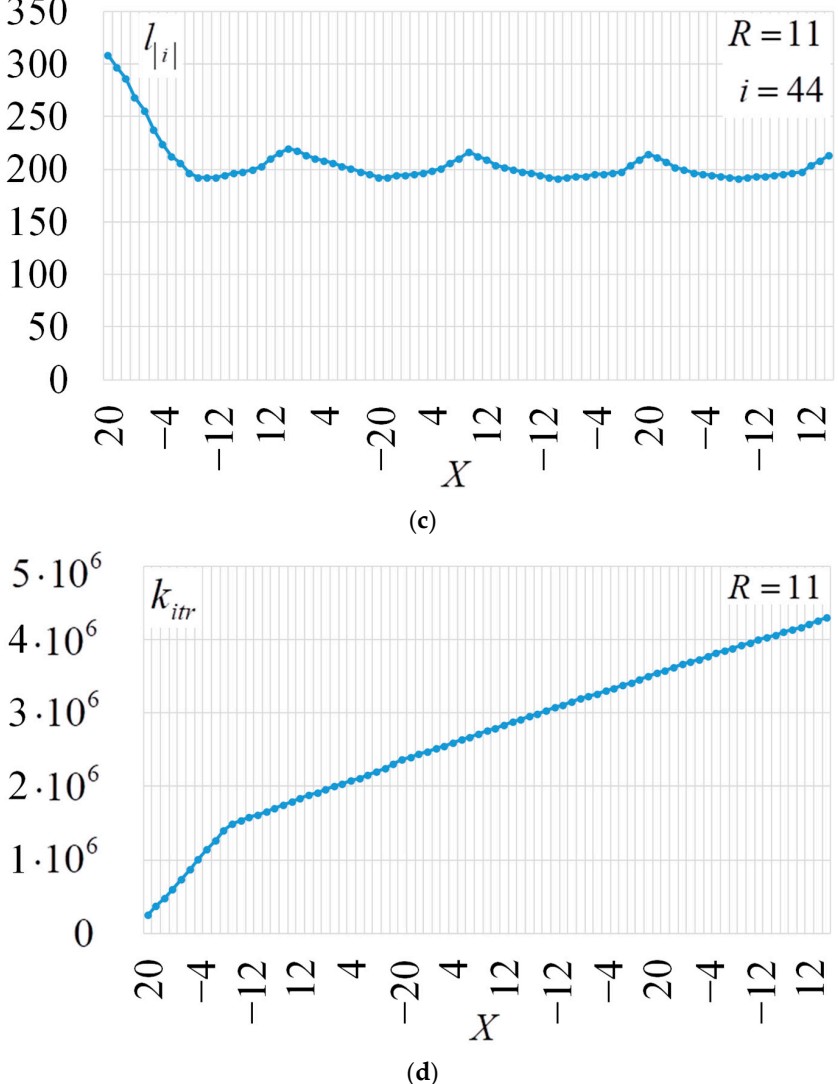

**Figure 8.** The trajectory of the $l_{|i|}$ S-box at dynamical change $X$ (4 cycles) and $R = 11$.

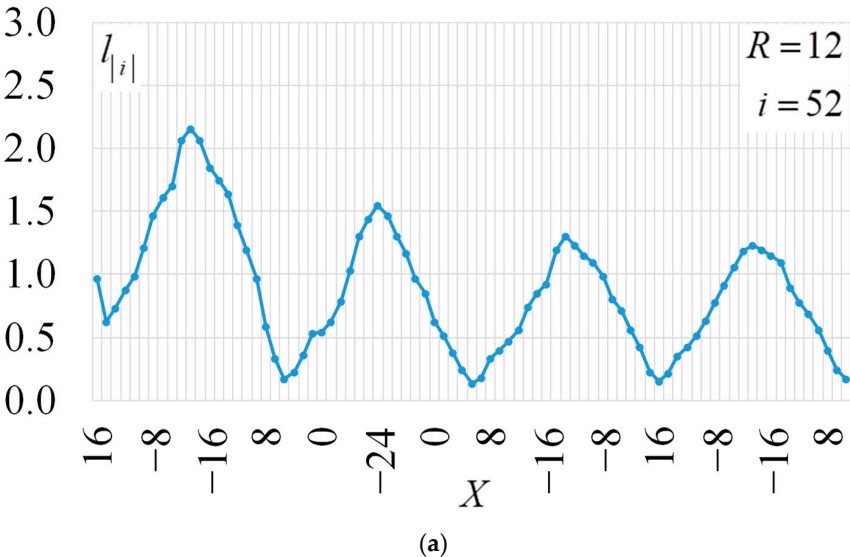

**Figure 9.** *Cont.*

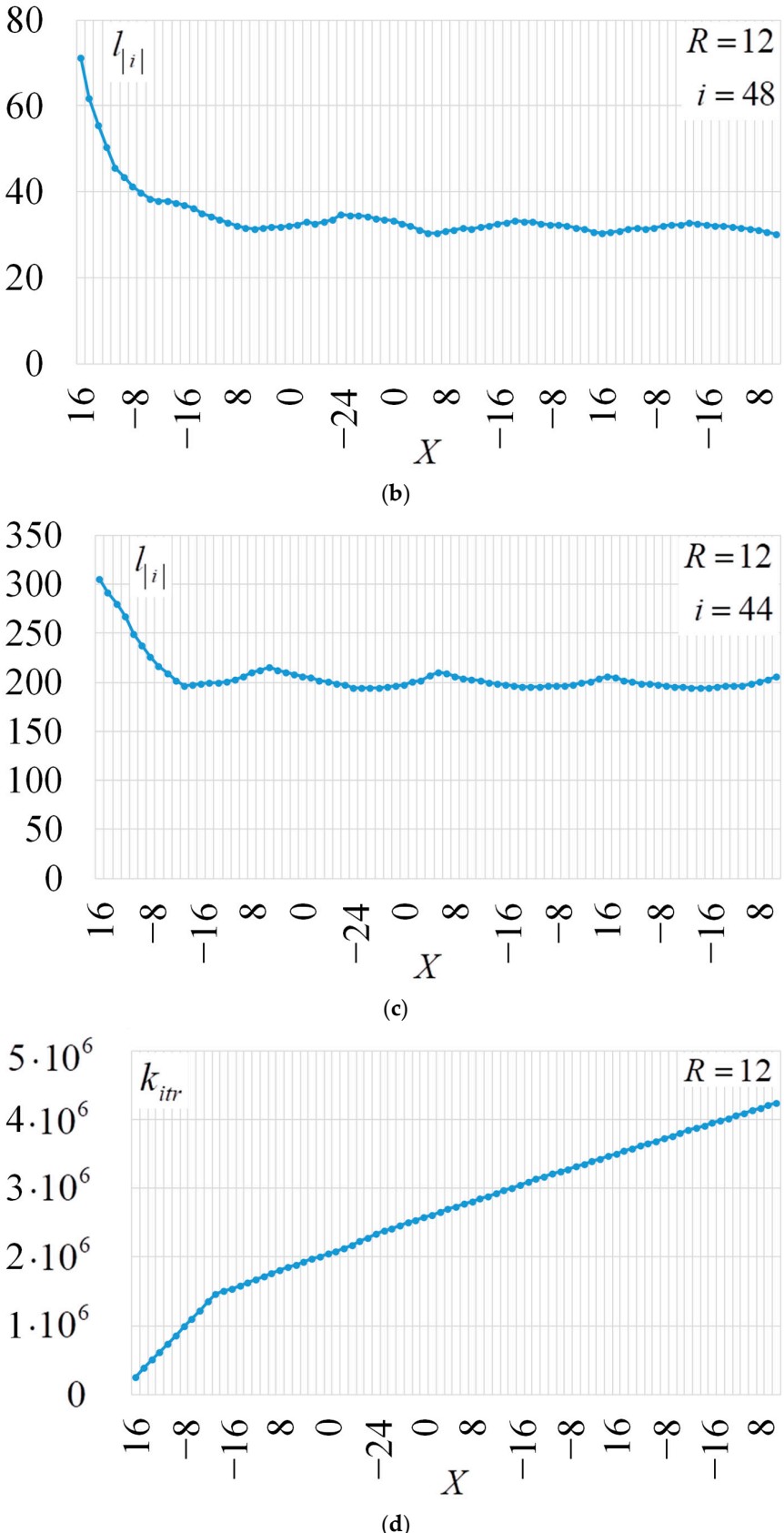

**Figure 9.** Trajectory of the $l_{|i|}$ S-box at dynamical change $X$ (4 cycles) and $R = 12$.

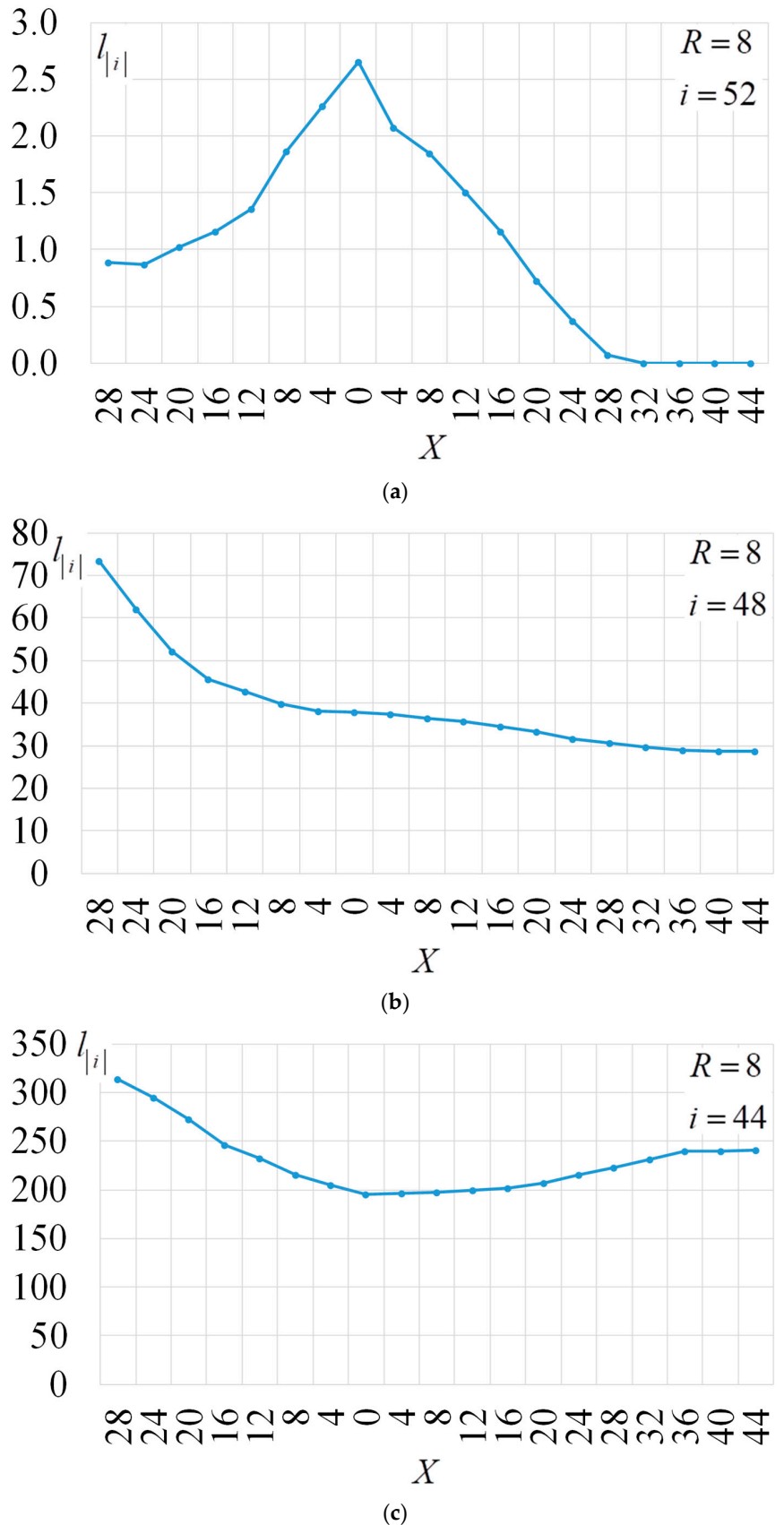

**Figure 10.** *Cont.*

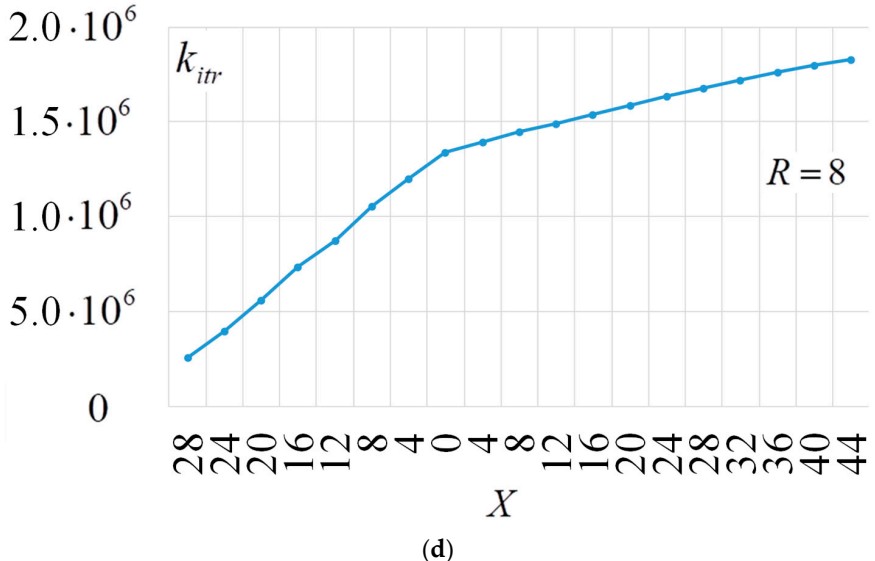

**(d)**

**Figure 10.** The trajectory of the $l_{|i|}$ S-box at dynamical change $X$ (1 cycles) and $R = 8$.

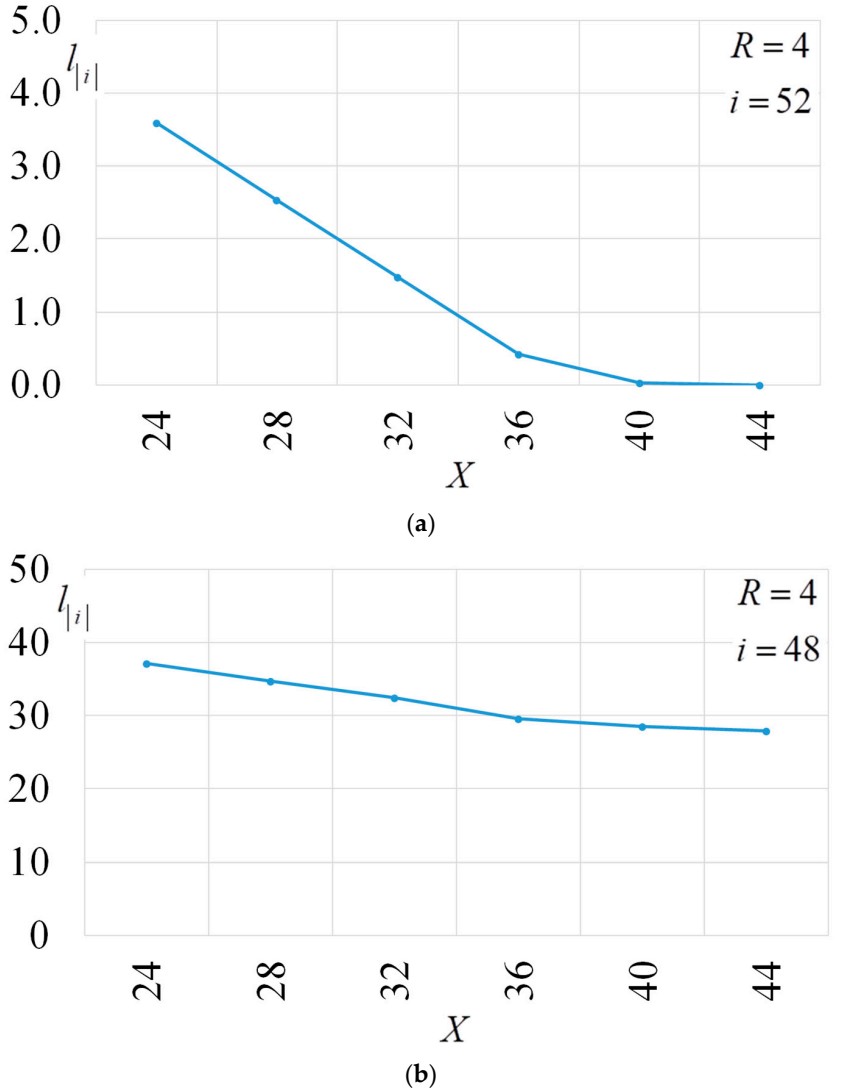

**(a)**

**(b)**

**Figure 11.** *Cont.*

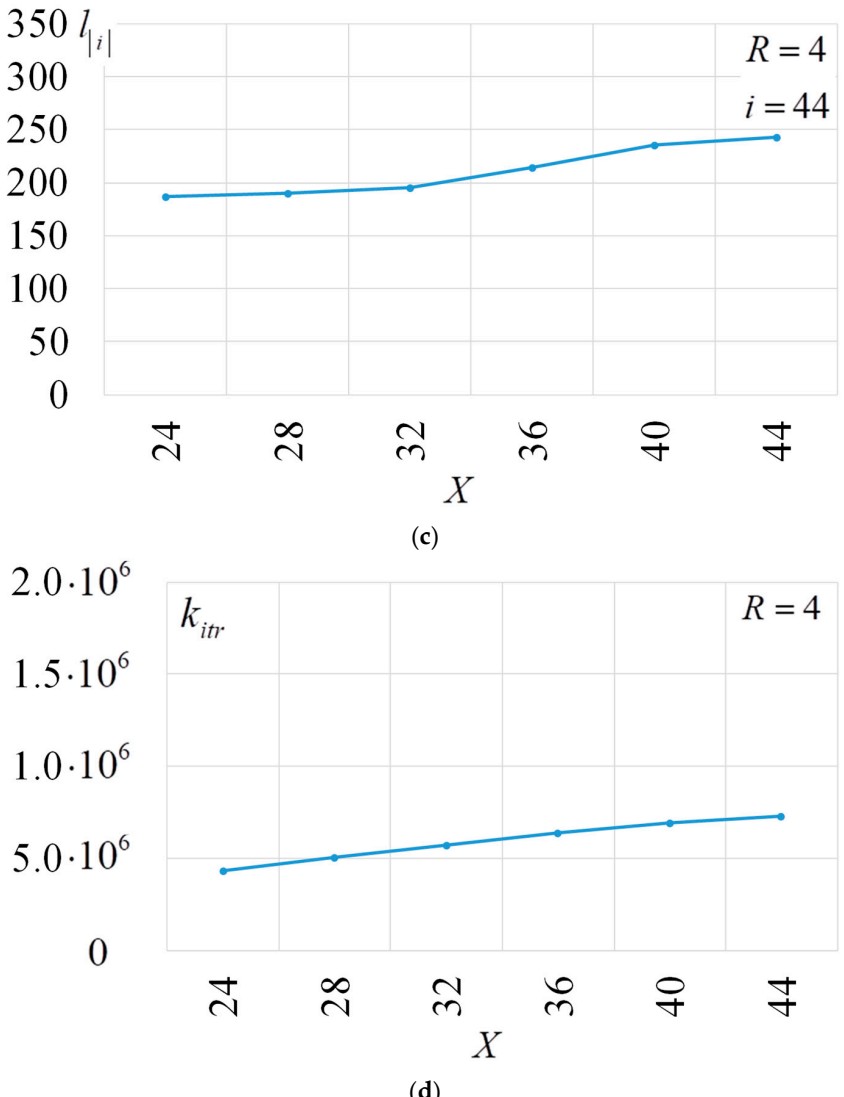

(c)

(d)

**Figure 11.** Trajectory of the $l_{|i|}$ S-box at dynamical reduction $X$ and $R = 4$.

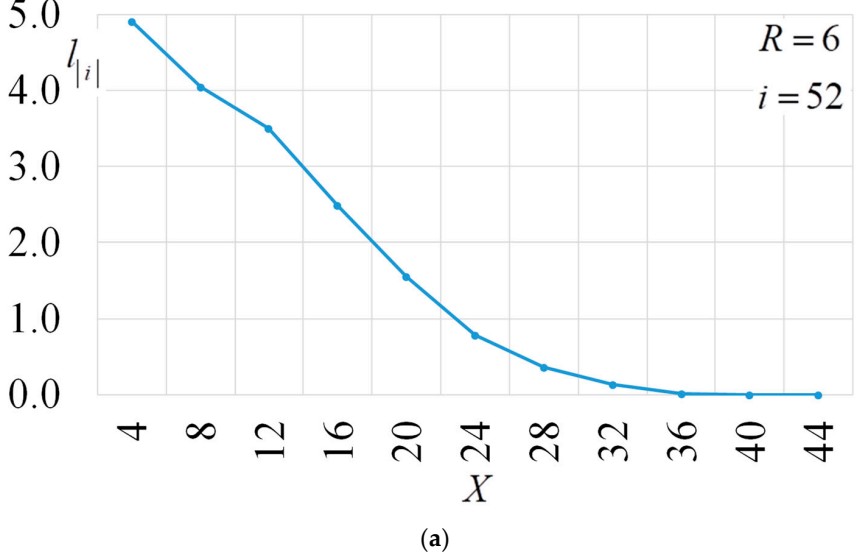

(a)

**Figure 12.** *Cont.*

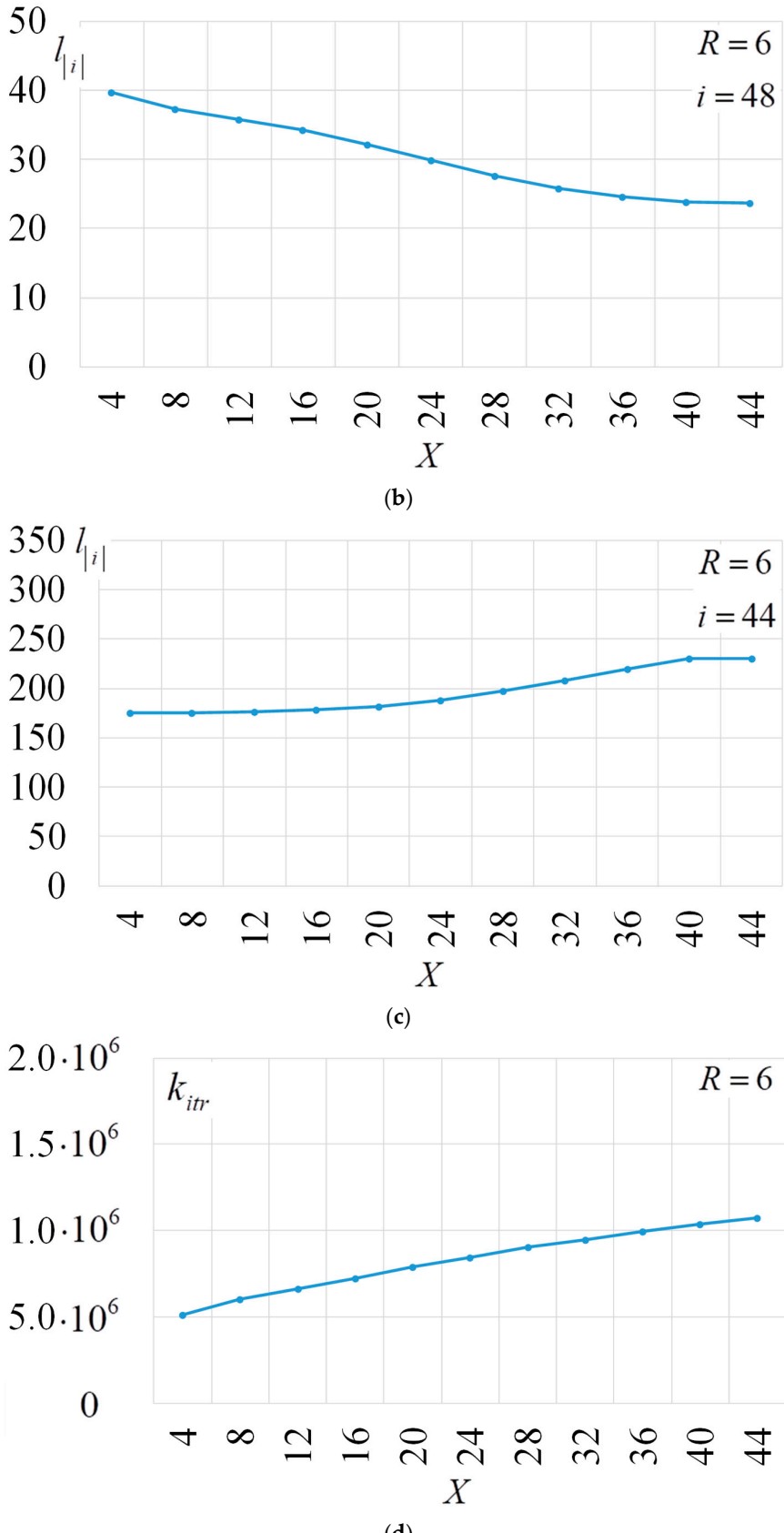

**Figure 12.** The trajectory of the $l_{|i|}$ S-box at dynamical reduction $X$ and $R = 6$.

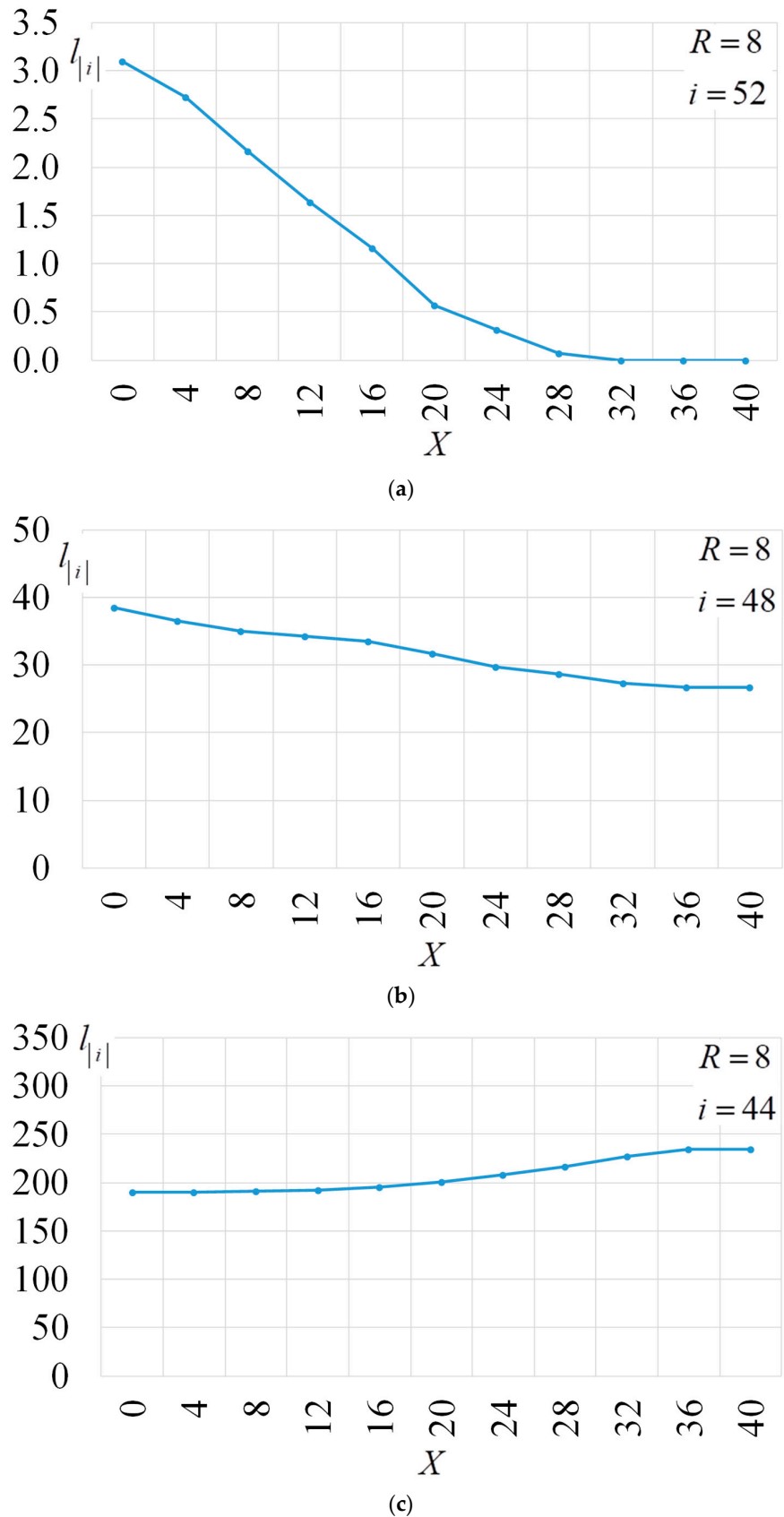

**Figure 13.** *Cont.*

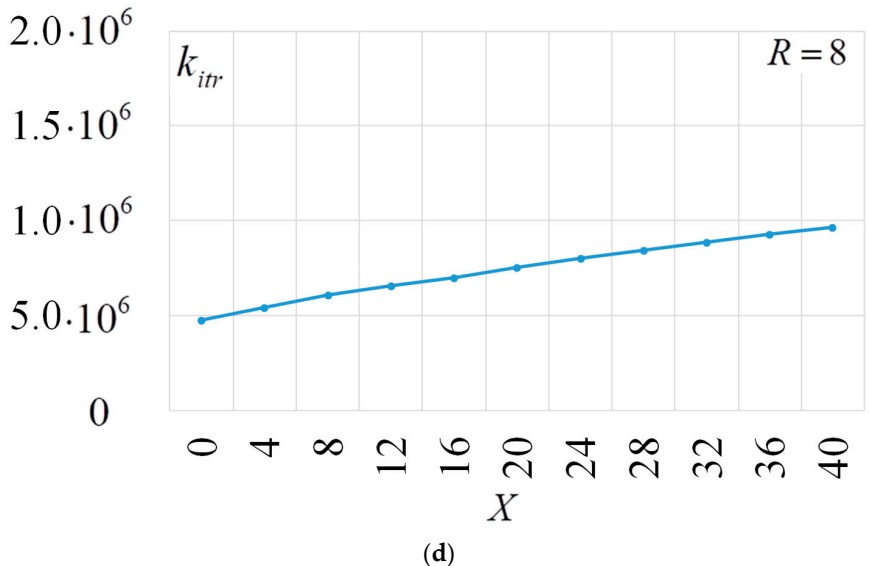

(**d**)

**Figure 13.** Trajectory of the $l_{|i|}$ S-box at dynamical reduction $X$ and $R = 8$.

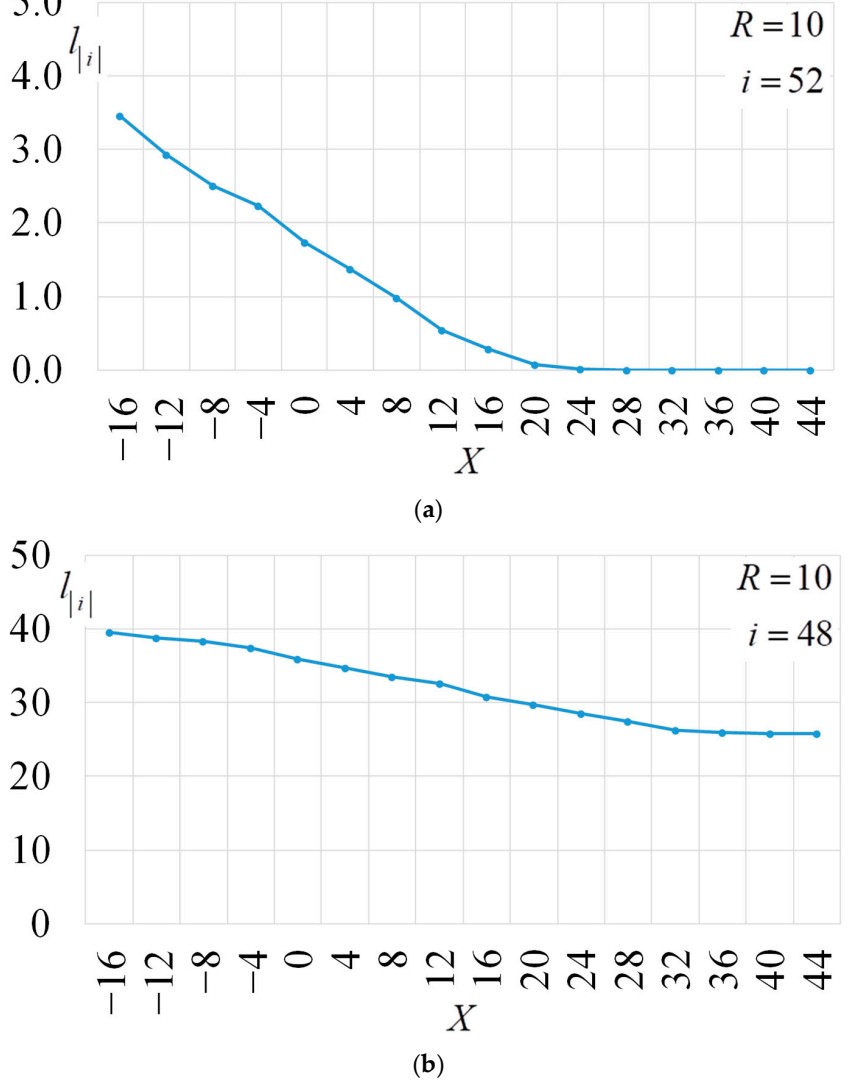

(**a**)

(**b**)

**Figure 14.** *Cont.*

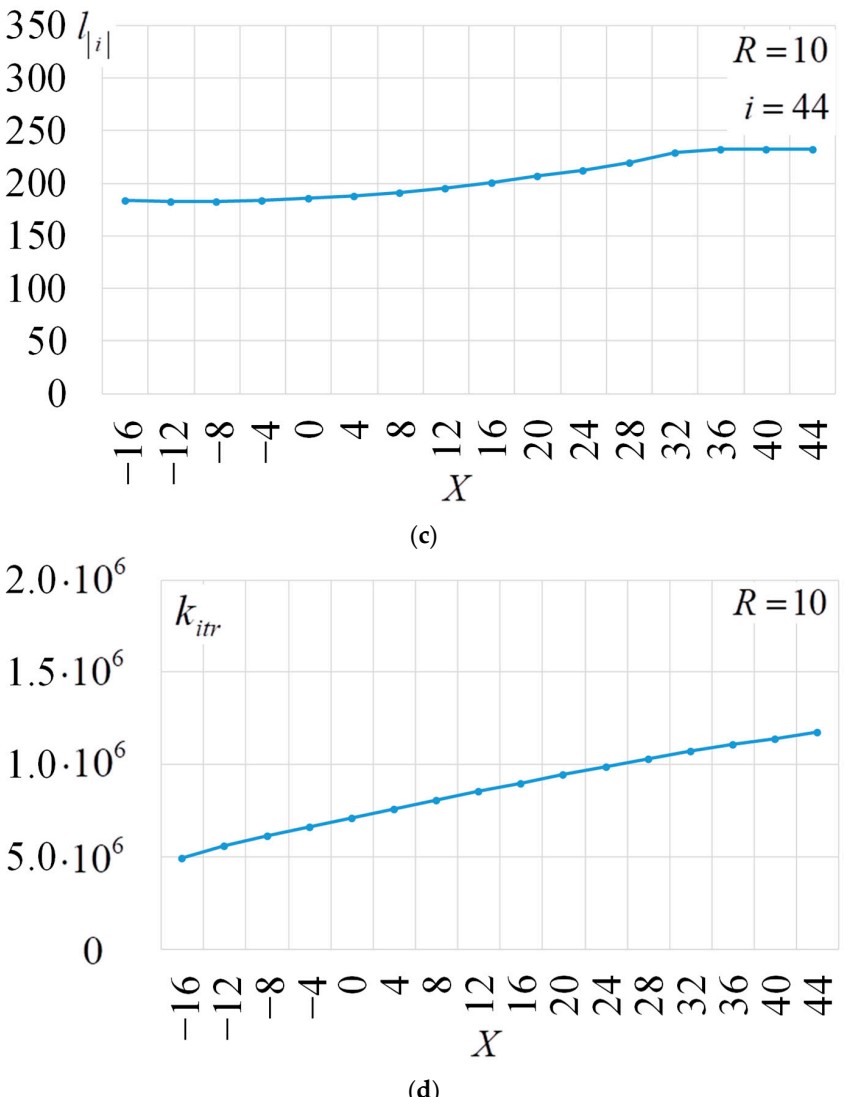

(c)

(d)

**Figure 14.** The trajectory of the $l_{|i|}$ S-box at dynamical reduction $X$ and $R = 10$.

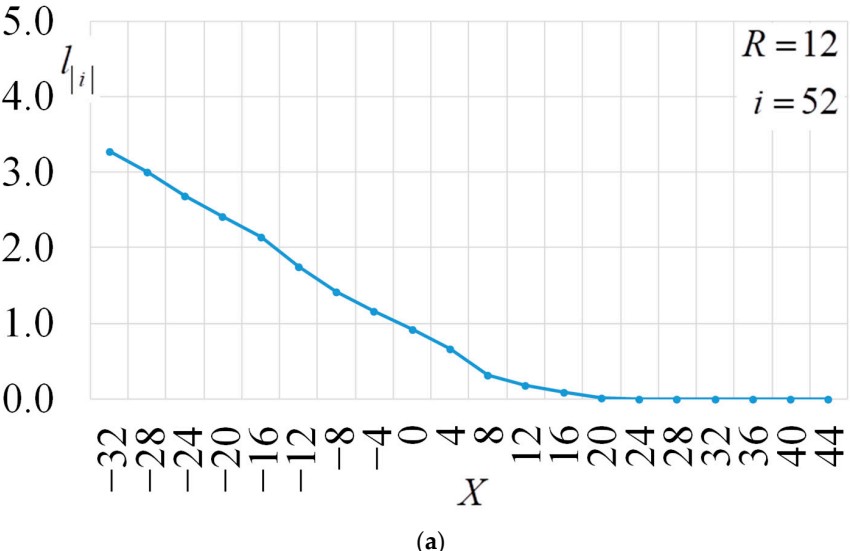

(a)

**Figure 15.** *Cont.*

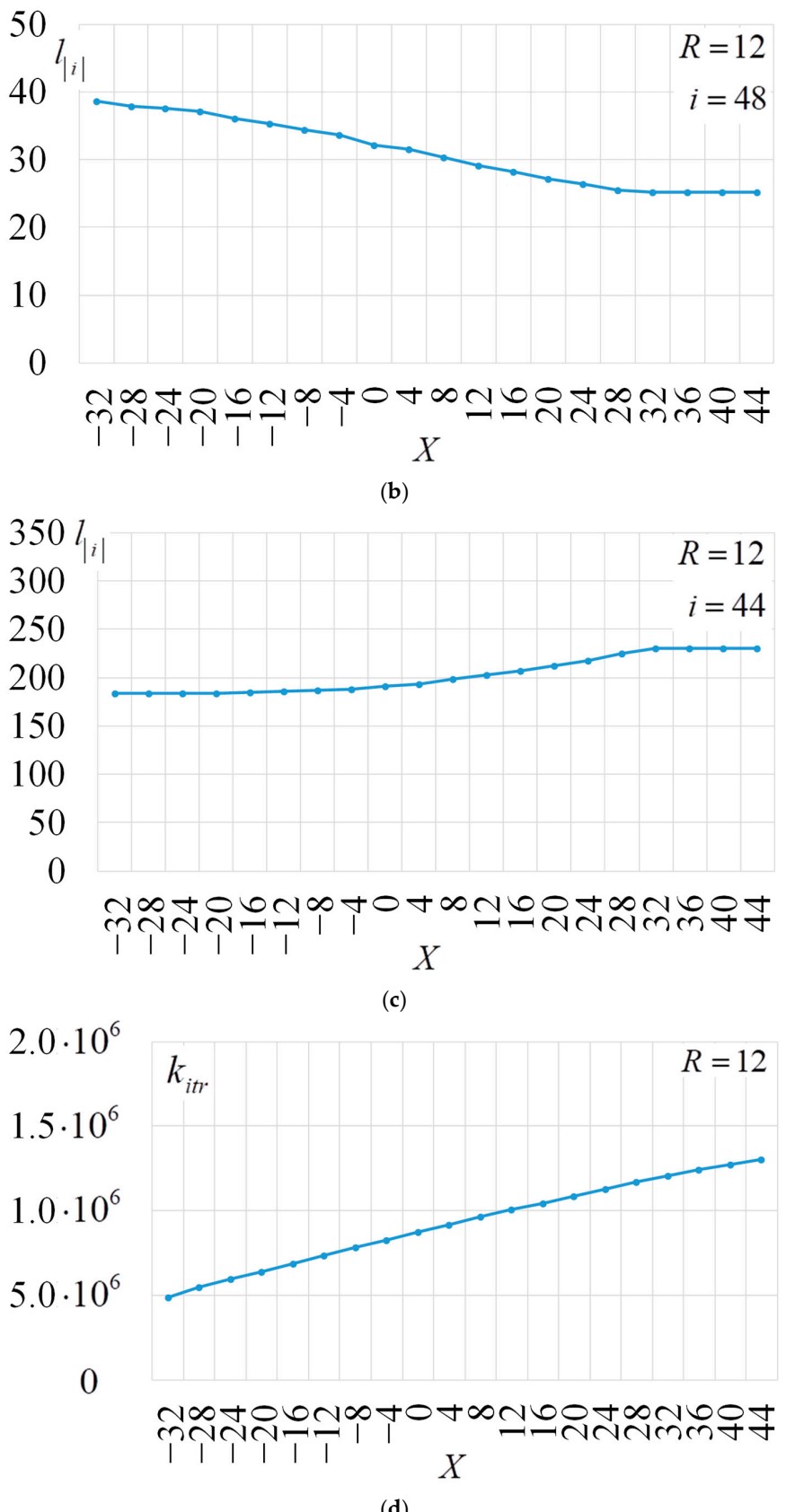

**Figure 15.** Trajectory of the $l_{|i|}$ S-box at dynamical reduction $X$ and $R = 12$.

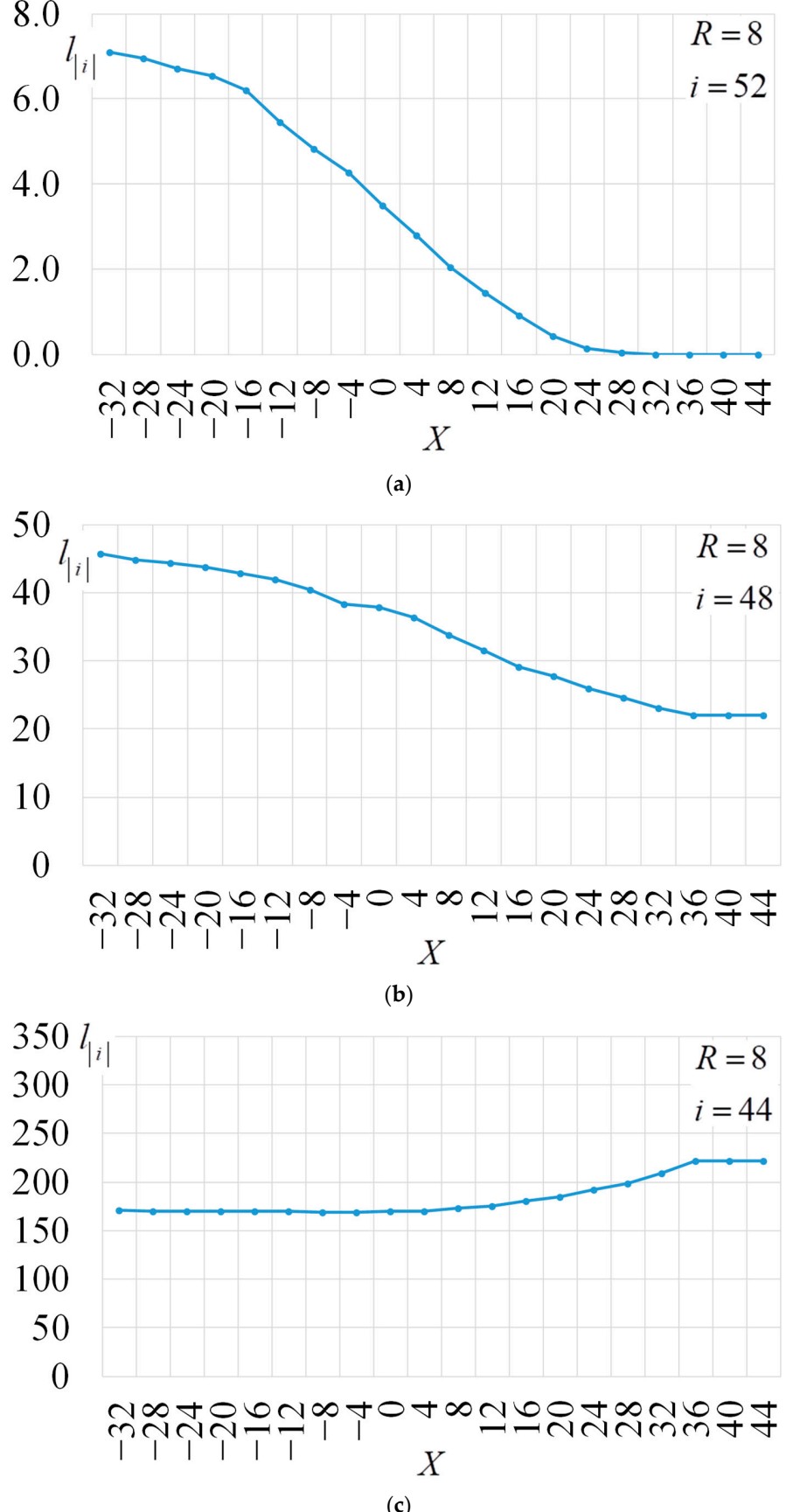

**Figure 16.** *Cont.*

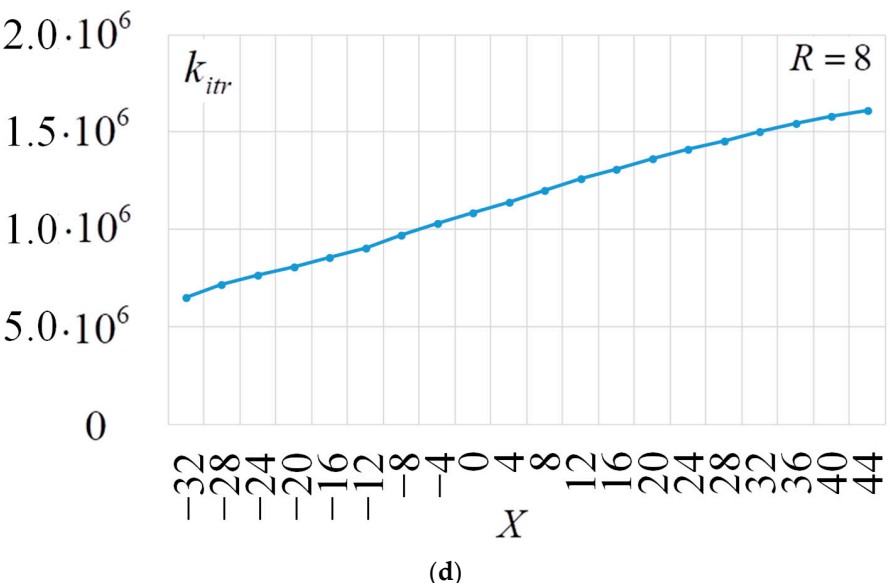

**(d)**

**Figure 16.** The trajectory of the $l_{|i|}$ S-box at dynamical reduction $X = -32 \div 4$ and $R = 8$.

- When applying the above dynamic changes in the parameter $X$, the search algorithm manages to almost halve the average value of the number of Walsh–Hadamard spectral coefficients in the position $i = 48$ from $l_{|i=48|} = 60 \div 75$ a fixed parameter $X$ to $l_{|i=48|} = 28 \div 31$; a cyclic change in the parameter $X$.

  However, about 10% of the runs still had coefficients at $i = 52$.

  At the end of each of the studied cycles, the values of $l_{|i=48|}$ and $l_{|i=52|}$ were in the minimum (for the cycle) state, and the value of $l_{|i=44|}$ was in the maximum state.

  Given that the value $l_{|i=48|}$ decreased with each of the four cycles, but not significantly (1–2 values), while each cycle significantly increased the number of iterations performed (at least 32,896 iterations were performed for each parameter $X$), it is advisable to limit this to one cycle. Taking into account the decrease in the values of $l_{|i=52|}$ with an increase in the parameter $l_{|i=52|}$, it is advisable to increase the parameter $X$ to the maximum value (for $N(S) = 106$ this, it will be $X = 44$). Taking into account the above considerations, the next 100 runs were carried out from one cycle and the parameter $X$ was changed in the following chain: for R = 8: X = 28; 24; 20; 16; 12; 8; 4; 0; 4; 8; 12; 16; 20; 24; 28; 32; 36; 40; 44.

  The results are shown in Figure 10.

  Starting from $X = 32$ and more, none of the values of $X$ had spectral coefficients greater than $i_{max} = 48$. The increase in the final parameter $X$ from 28 to 44, on average, led to a slight decrease (by onr coefficient) in the average statistical value of $l_{|i=48|}$. Using one cycle instead of four or five allowed us to reduce the average statistical number of iterations by more than half (to 1,830,951 iterations).

  Our next step was to reduce the cycle by starting it from the minimum values of $X$. For all runs of the search algorithm, the value of the parameter $R$ was also recorded. The tests were performed with R = 4; 6; 8; 10; 12. The results were averaged over 100 runs. In each run, a cycle of parameter $X$ changes was performed in the following chains (see Table 3).

**Table 3.** Selected parameters X and R in the first series of experiments (the second part).

| R | | | | | | X | | | | | | | | | | | | | | |
|---|---|---|---|---|---|---|---|---|---|---|---|---|---|---|---|---|---|---|---|---|
| 4 | 24 | 28 | 32 | 36 | 40 | 44 | | | | | | | | | | | | | | |
| 6 | 4 | 8 | 12 | 16 | 20 | 24 | 28 | 32 | 36 | 40 | 44 | | | | | | | | | |
| 8 | 0 | 4 | 8 | 12 | 16 | 20 | 24 | 28 | 32 | 36 | 40 | 44 | | | | | | | | |
| 10 | −16 | −12 | −8 | −4 | 0 | 4 | 8 | 12 | 16 | 20 | 24 | 28 | 32 | 36 | 40 | 44 | | | | |
| 12 | −32 | −28 | −24 | −20 | −16 | −12 | −8 | −4 | 0 | 4 | 8 | 12 | 16 | 20 | 24 | 28 | 32 | 36 | 40 | 44 |

The results are shown in Figures 11–15. Reducing the cycle only to the region where the parameter $X$ was reduced made it possible to reduce the average statistical value by four, to $l_{|i=48|} = 24 \div 27$ (at all $l_{|i=52|} = 0$). Expanding the range of changes in values

$X = -32; -28; -24; -20; -16; -12; -8; -4; 0; 4; 8; 12; 16; 20; 24; 28; 32; 36; 40; 44$

with a fixed value R = 8 led to an even greater average statistical decrease $l_{|i=48|} = 22$. The results are shown in Figure 16.

When commenting on the experimental results, it should be noted that the probability of finding an improvement in the objective function slightly decreases with an increase in the parameter $X$. However, as the change in the parameter $X$ increases, the weight values of the coefficients in the spectrum distribution change. This leads to a corresponding calculation of the objective function. As a result, it becomes easier to find the improvement of the objective function. This result can be used to optimize the search.

### 5.5.2. Exploring the Interplay of Cost Function Parameters

The second series of experiments focuses on the interplay between the cost function parameters R and X. By systematically varying these parameters and examining the resulting performance metrics, we uncover the optimal parameter combinations that lead to an improved efficiency and higher nonlinearity in the generated S-boxes.

These tests were carried out with a slight change in the parameter $R$ (up to three units), while the parameter $X$ was changed in such a way that the changes for each slight change $R$ were parallel to the values determined by the ratio $X = 52 - 4 \cdot R$. The results were averaged over 100 runs. In each run, a cycle of parameters $R$ and $X$ changes was performed according to the chain shown in Table 4. The results are shown in Figures 17–22 (the value of the change in $R$ is not shown in the figure).

**Table 4.** Changes in parameters $R$ and $X$ in the search algorithm runs, the results of which are shown in Figures 17–22.

|  | $R \in [2, 3, 4]$ | $R \in [4, 5, 6]$ | $R \in [6, 7, 8]$ | $R \in [8, 9, 10]$ | $R \in [10, 11, 12]$ | $R \in [12, 13, 14]$ | $X \in [-20 \ldots 40]$ |
|---|---|---|---|---|---|---|---|
| 0 | 2 | 4 | 6 | 8 | 10 | 12 | −20 |
| 1 | 2 | 4 | 6 | 8 | 10 | 12 | −16 |
| 2 | 3 | 5 | 7 | 9 | 11 | 13 | −20 |
| 3 | 4 | 6 | 8 | 10 | 12 | 14 | −24 |
| 4 | 2 | 4 | 6 | 8 | 10 | 12 | −12 |
| 5 | 3 | 5 | 7 | 9 | 11 | 13 | −16 |
| 6 | 4 | 6 | 8 | 10 | 12 | 14 | −20 |
| 7 | 2 | 4 | 6 | 8 | 10 | 12 | −8 |
| 8 | 3 | 5 | 7 | 9 | 11 | 13 | −12 |
| 9 | 4 | 6 | 8 | 10 | 12 | 14 | −16 |
| 10 | 2 | 4 | 6 | 8 | 10 | 12 | −4 |
| 11 | 3 | 5 | 7 | 9 | 11 | 13 | −8 |
| 12 | 4 | 6 | 8 | 10 | 12 | 14 | −12 |
| 13 | 2 | 4 | 6 | 8 | 10 | 12 | 0 |
| 14 | 3 | 5 | 7 | 9 | 11 | 13 | −4 |
| 15 | 4 | 6 | 8 | 10 | 12 | 14 | −8 |
| 16 | 2 | 4 | 6 | 8 | 10 | 12 | 4 |
| 17 | 3 | 5 | 7 | 9 | 11 | 13 | 0 |
| 18 | 4 | 6 | 8 | 10 | 12 | 14 | −4 |
| 19 | 2 | 4 | 6 | 8 | 10 | 12 | 8 |
| 20 | 3 | 5 | 7 | 9 | 11 | 13 | 4 |
| 21 | 4 | 6 | 8 | 10 | 12 | 14 | 0 |

**Table 4.** *Cont.*

|  | R ∈ [2, 3, 4] | R ∈ [4, 5, 6] | R ∈ [6, 7, 8] | R ∈ [8, 9, 10] | R ∈ [10, 11, 12] | R ∈ [12, 13, 14] | X ∈ [−20 . . . 40] |
|---|---|---|---|---|---|---|---|
| 22 | 2 | 4 | 6 | 8 | 10 | 12 | 12 |
| 23 | 3 | 5 | 7 | 9 | 11 | 13 | 8 |
| 24 | 4 | 6 | 8 | 10 | 12 | 14 | 4 |
| 25 | 2 | 4 | 6 | 8 | 10 | 12 | 16 |
| 26 | 3 | 5 | 7 | 9 | 11 | 13 | 12 |
| 27 | 4 | 6 | 8 | 10 | 12 | 14 | 8 |
| 28 | 2 | 4 | 6 | 8 | 10 | 12 | 20 |
| 29 | 3 | 5 | 7 | 9 | 11 | 13 | 16 |
| 30 | 4 | 6 | 8 | 10 | 12 | 14 | 12 |
| 31 | 2 | 4 | 6 | 8 | 10 | 12 | 24 |
| 32 | 3 | 5 | 7 | 9 | 11 | 13 | 20 |
| 33 | 4 | 6 | 8 | 10 | 12 | 14 | 16 |
| 34 | 2 | 4 | 6 | 8 | 10 | 12 | 28 |
| 35 | 3 | 5 | 7 | 9 | 11 | 13 | 24 |
| 36 | 4 | 6 | 8 | 10 | 12 | 14 | 20 |
| 37 | 2 | 4 | 6 | 8 | 10 | 12 | 32 |
| 38 | 3 | 5 | 7 | 9 | 11 | 13 | 28 |
| 39 | 4 | 6 | 8 | 10 | 12 | 14 | 24 |
| 40 | 2 | 4 | 6 | 8 | 10 | 12 | 36 |
| 41 | 3 | 5 | 7 | 9 | 11 | 13 | 32 |
| 42 | 4 | 6 | 8 | 10 | 12 | 14 | 28 |
| 43 | 2 | 4 | 6 | 8 | 10 | 12 | 40 |
| 44 | 3 | 5 | 7 | 9 | 11 | 13 | 36 |
| 45 | 4 | 6 | 8 | 10 | 12 | 14 | 32 |
| 46 | 2 | 4 | 6 | 8 | 10 | 12 | 44 |
| 47 | 3 | 5 | 7 | 9 | 11 | 13 | 40 |
| 48 | 4 | 6 | 8 | 10 | 12 | 14 | 36 |

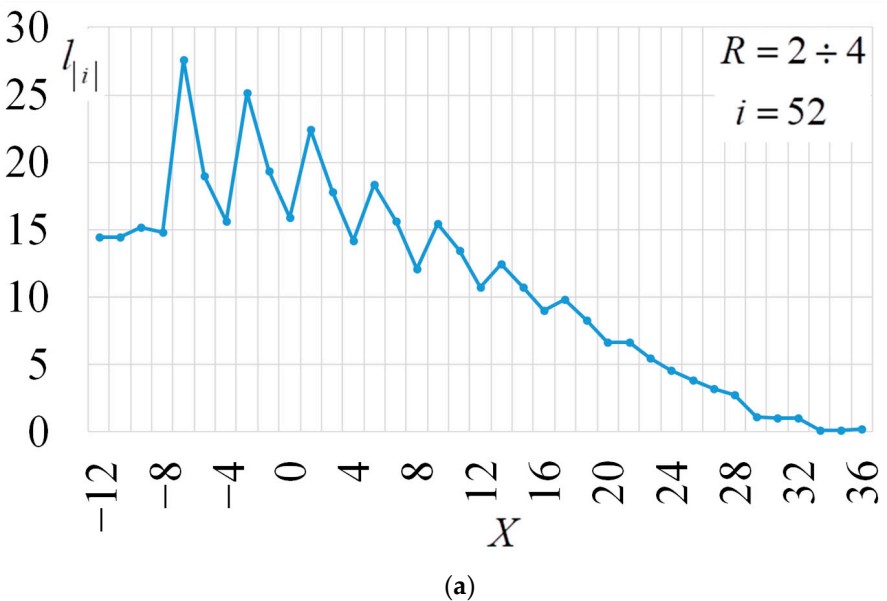

(**a**)

**Figure 17.** *Cont.*

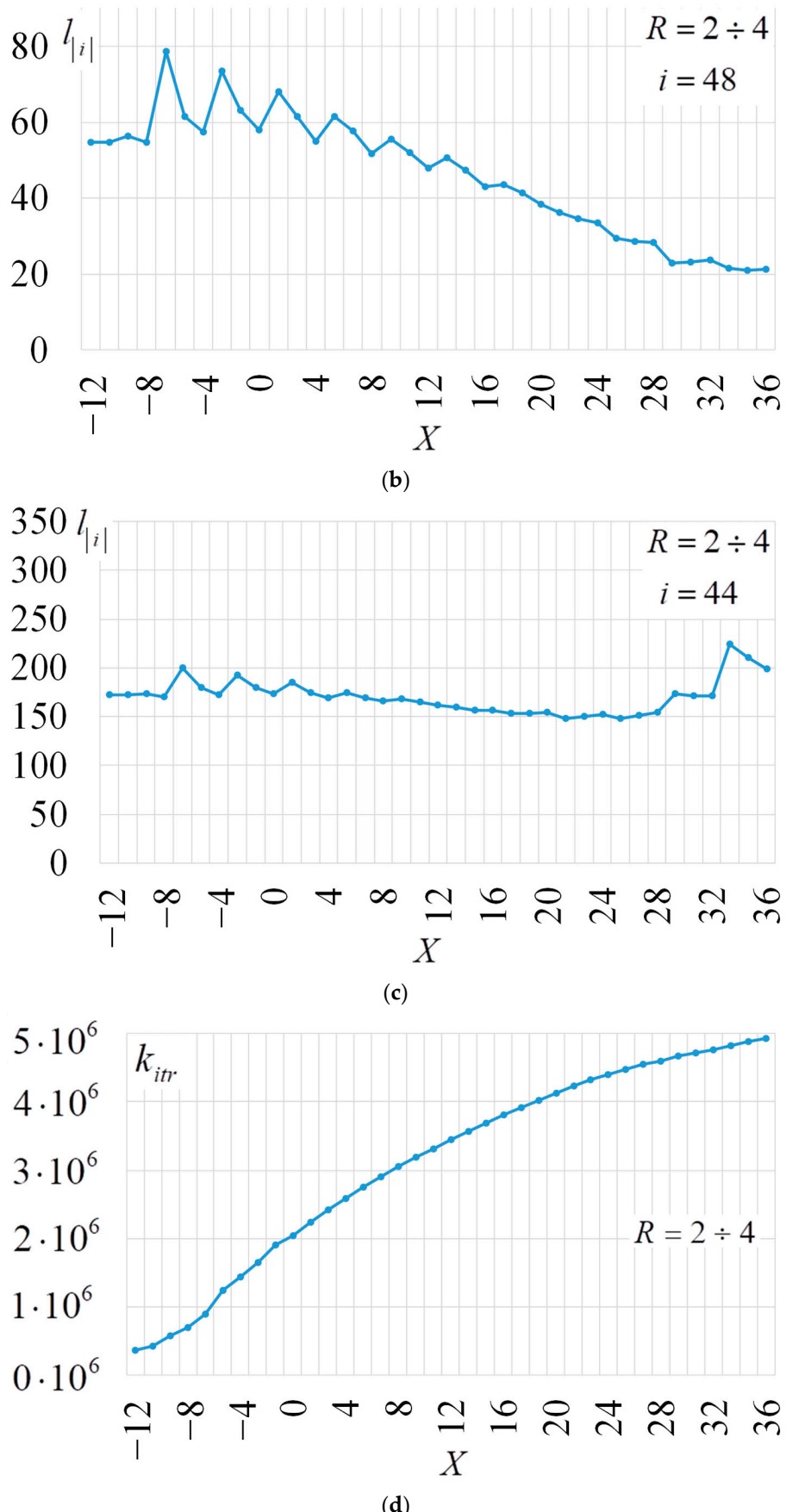

**Figure 17.** Trajectory of the $l_{|i|}$ S-box at dynamical change of parameters for $R = 2 \div 4$.

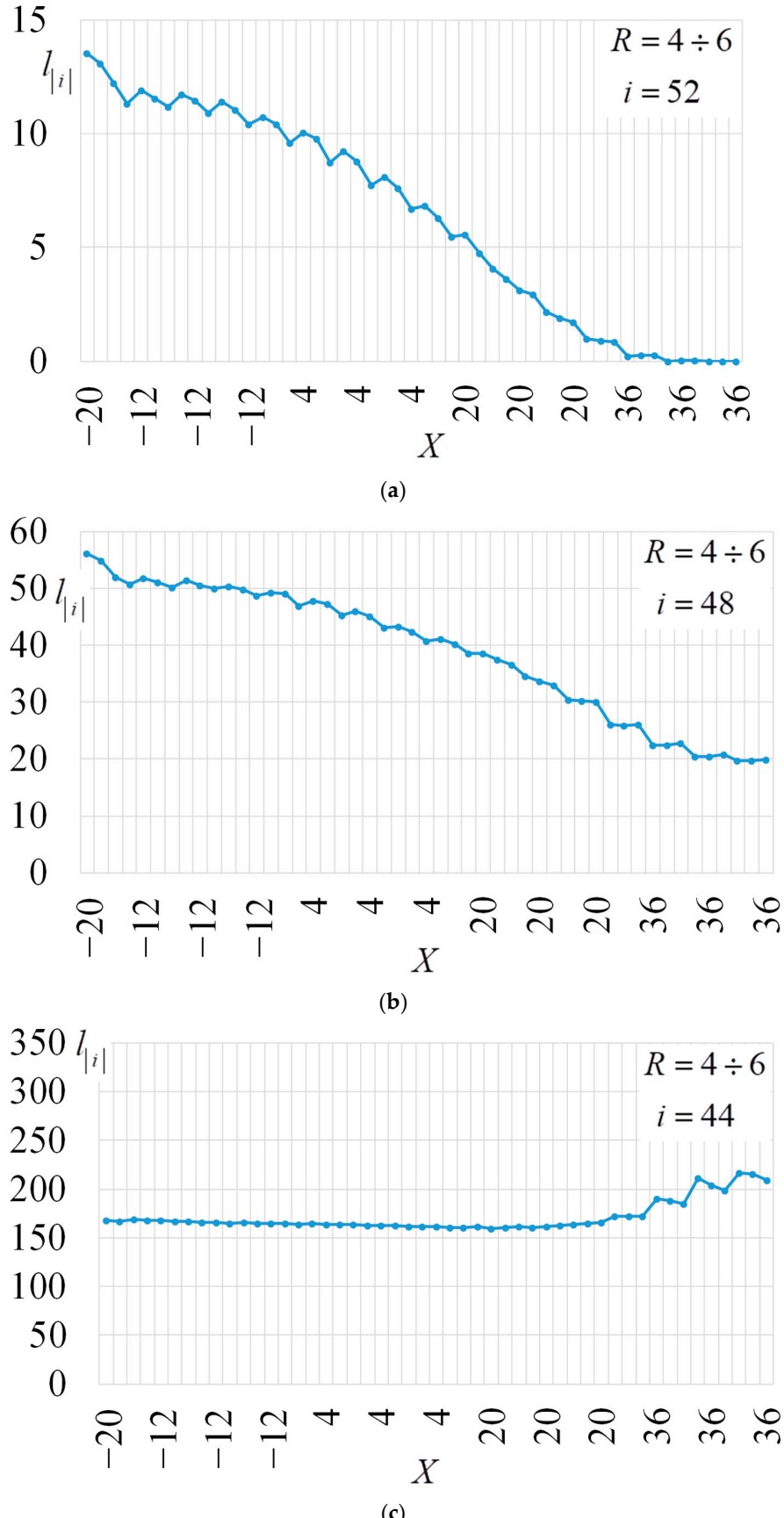

**Figure 18.** *Cont.*

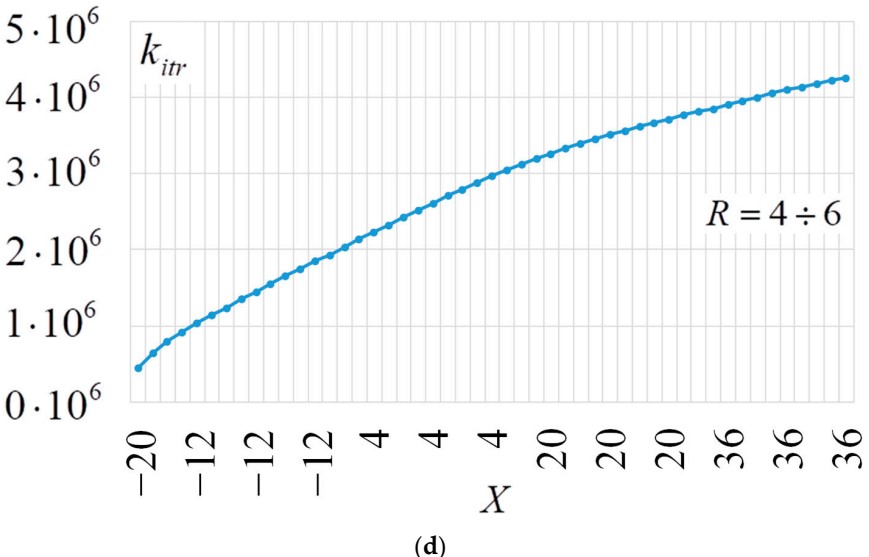

(**d**)

**Figure 18.** Trajectory of the $l_{|i|}$ S-box at dynamical change of parameters for $R = 4 \div 6$.

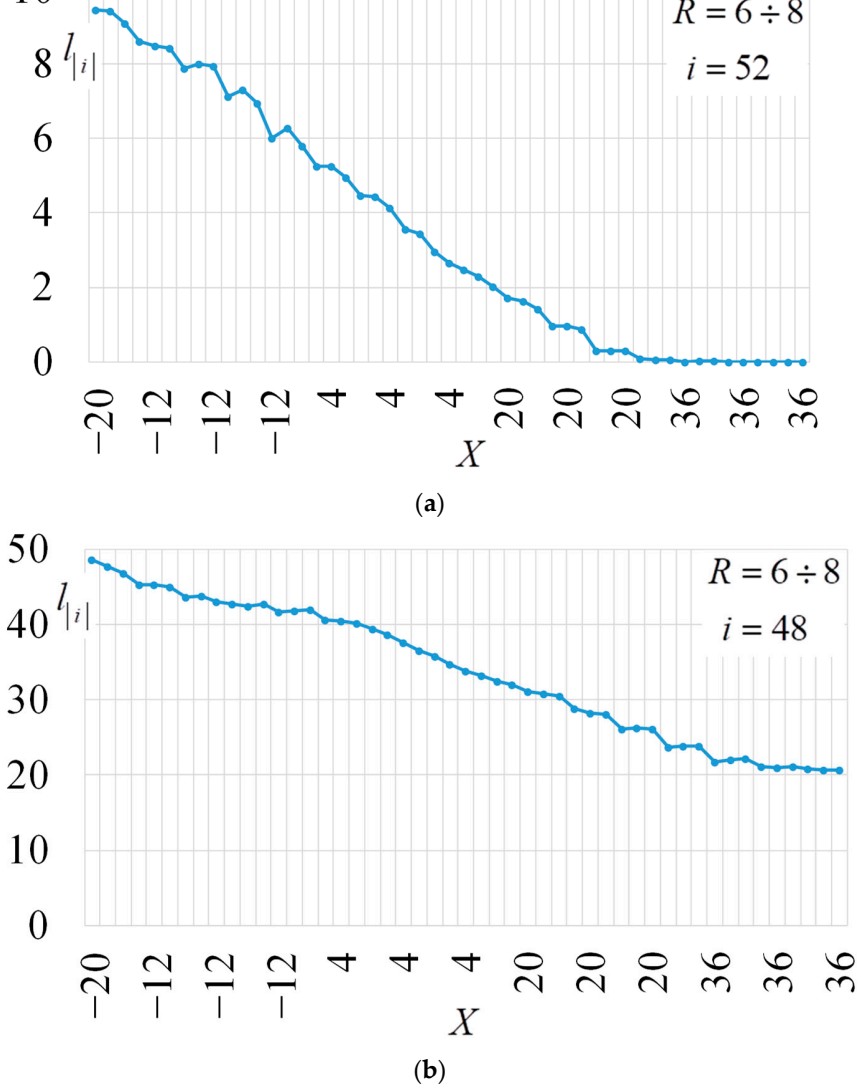

(**a**)

(**b**)

**Figure 19.** *Cont.*

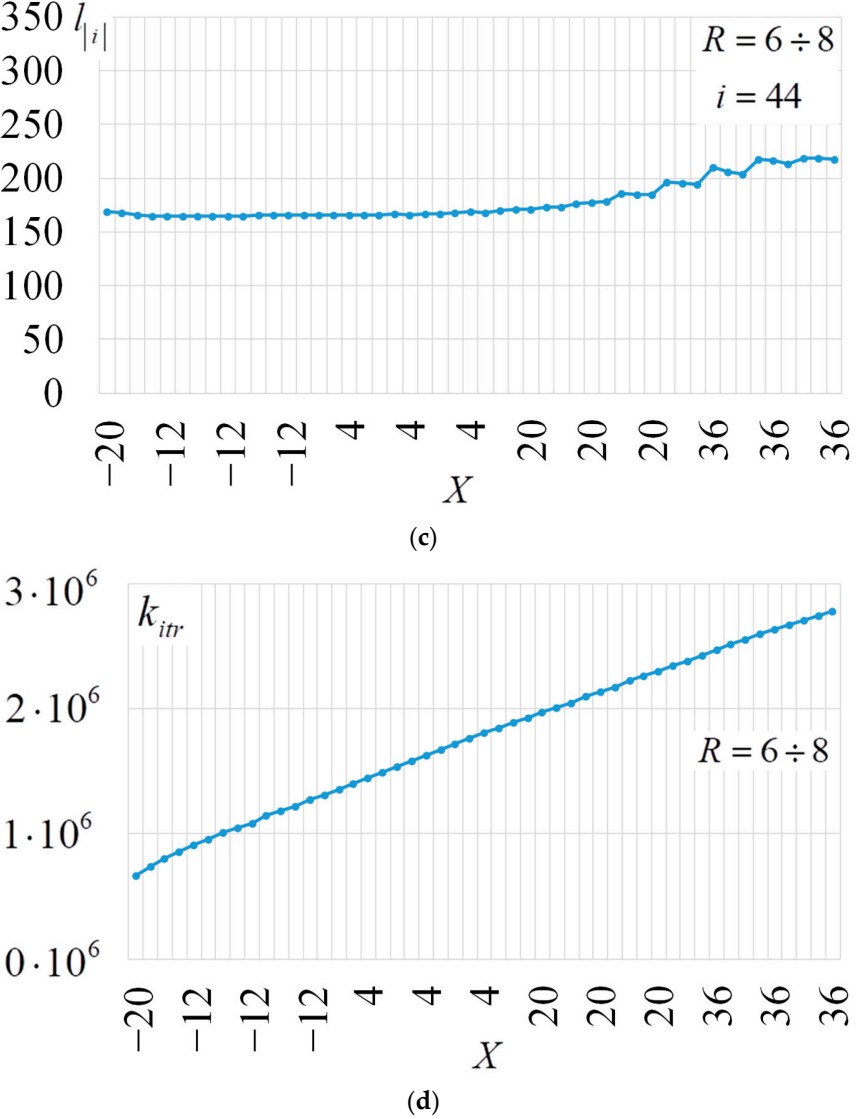

(c)

(d)

**Figure 19.** Trajectory of the $l_{|i|}$ S-box at dynamical change of parameters for $R = 6 \div 8$.

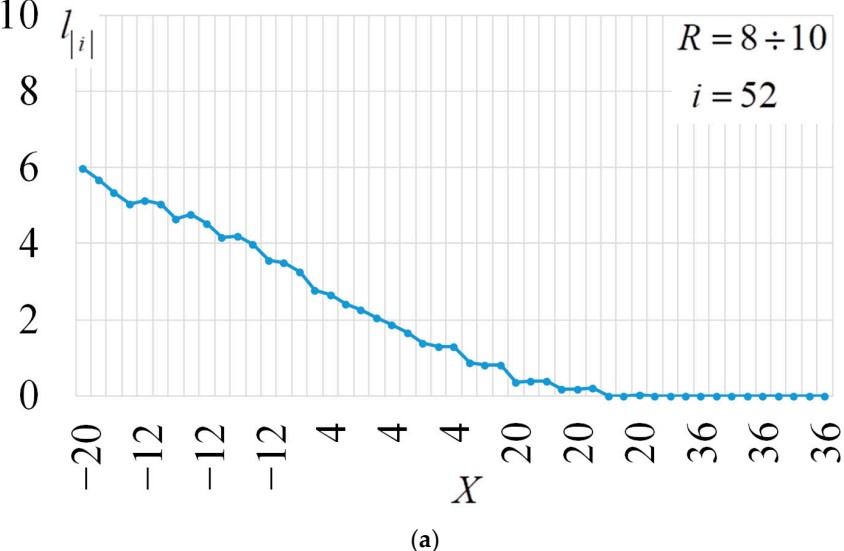

(a)

**Figure 20.** *Cont.*

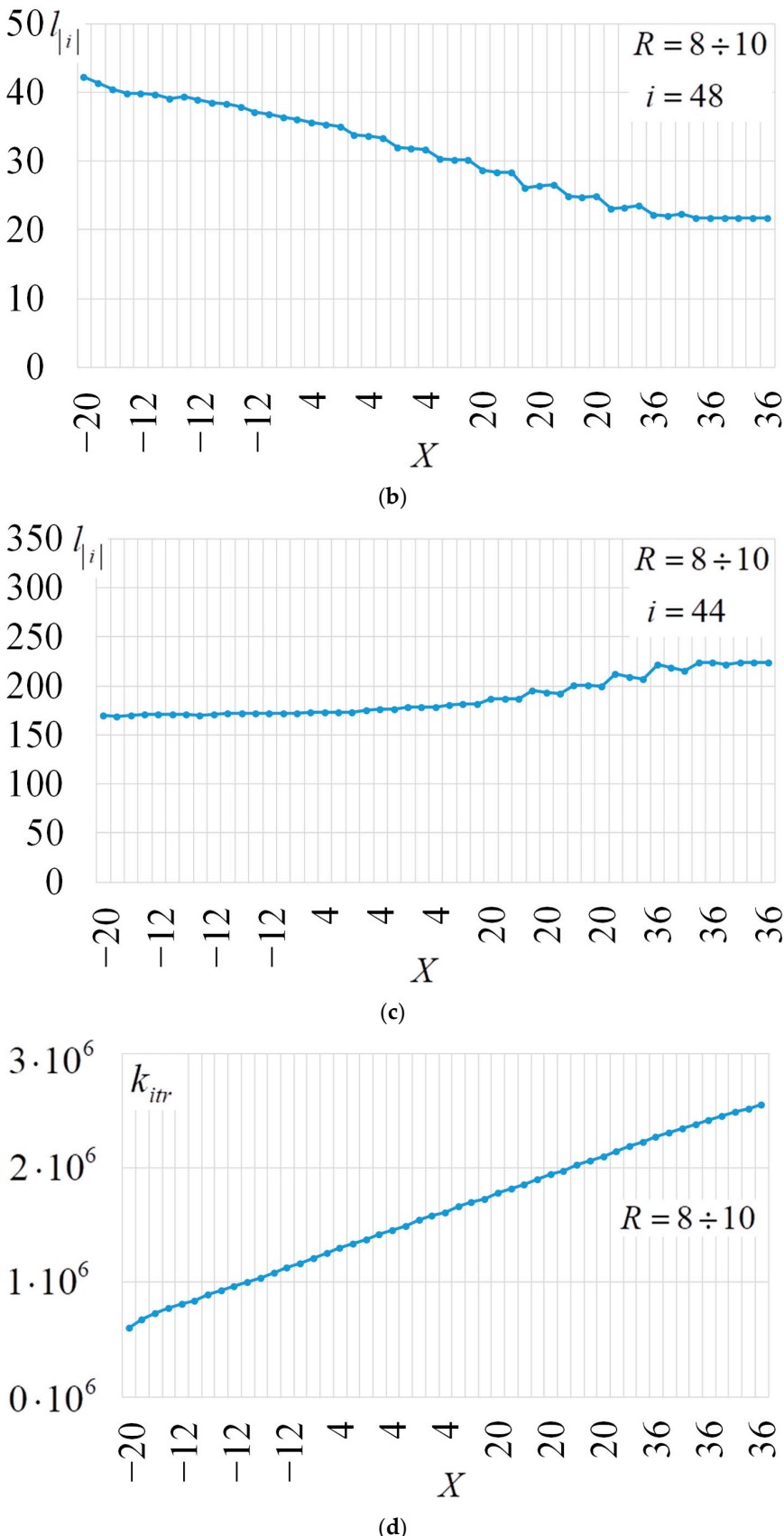

**Figure 20.** Trajectory of the $l_{|i|}$ S-box at dynamical change of parameters for $R = 8 \div 10$.

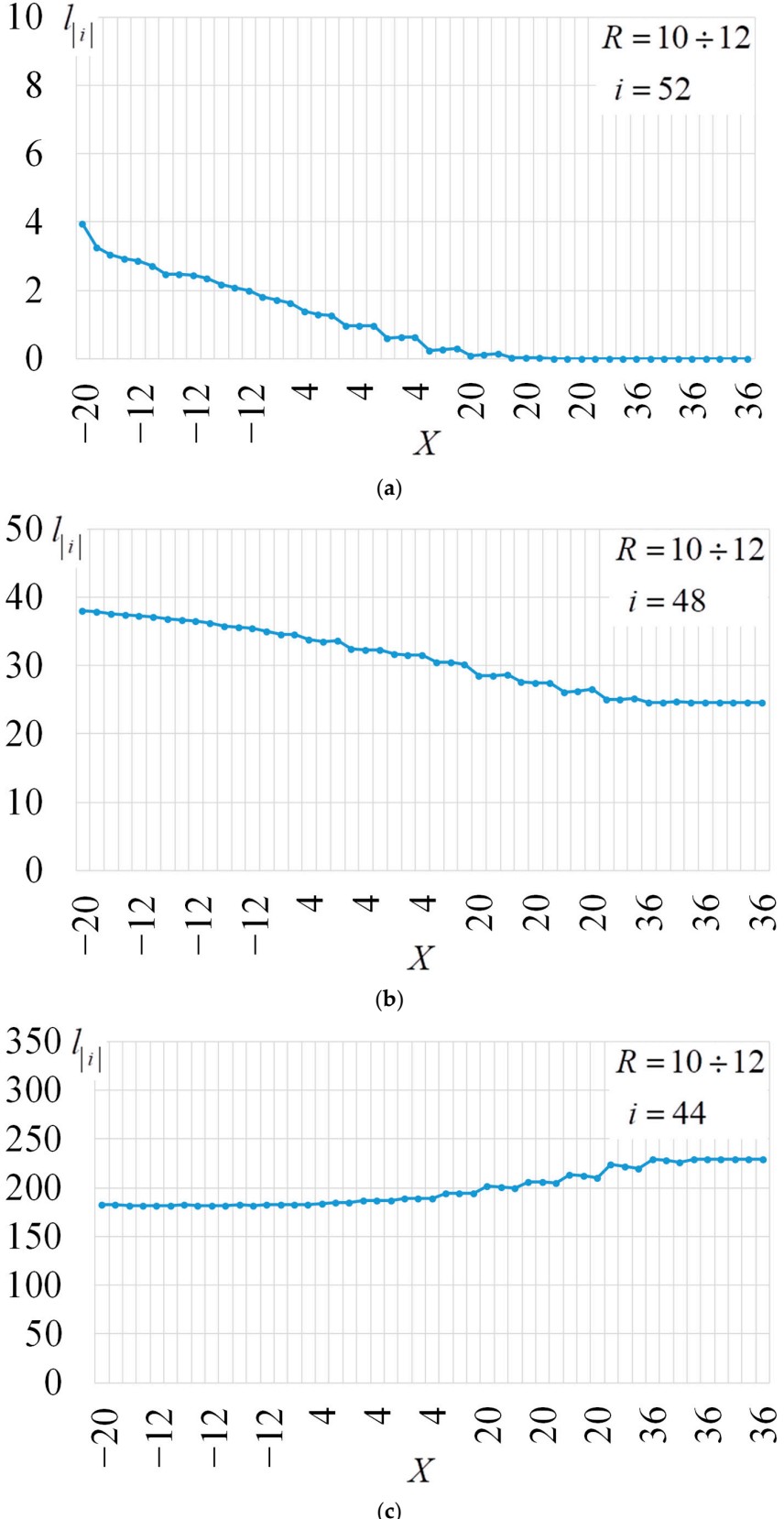

**Figure 21.** *Cont.*

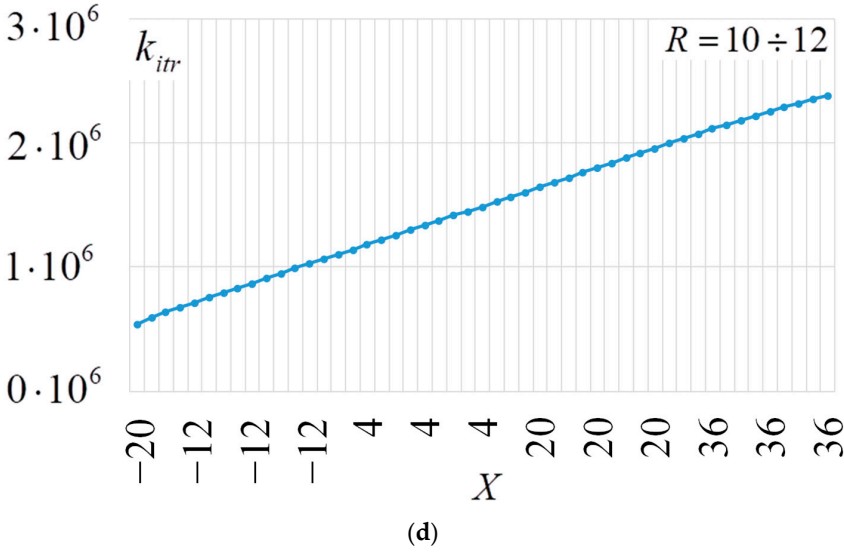

**Figure 21.** Trajectory of the $l_{|i|}$ S-box at dynamical change of parameters for $R = 10 \div 12$.

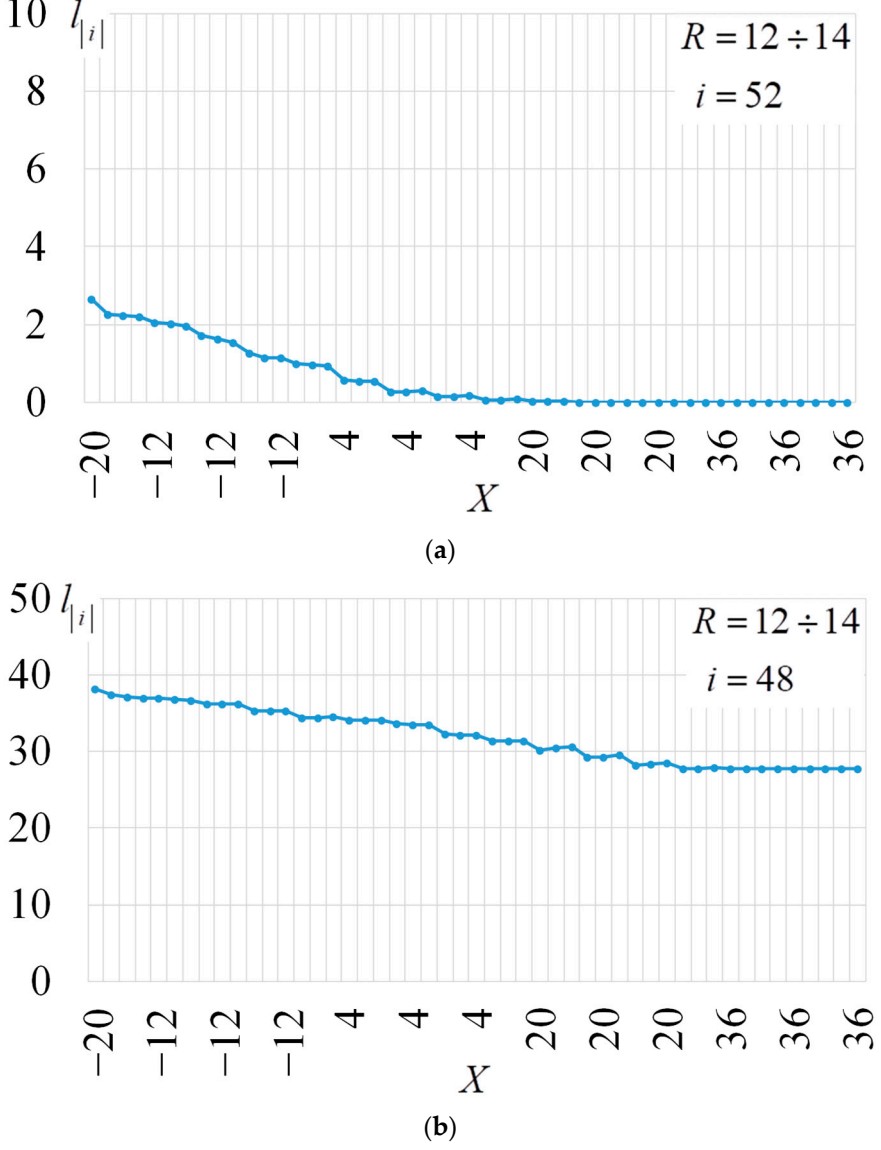

**Figure 22.** *Cont.*

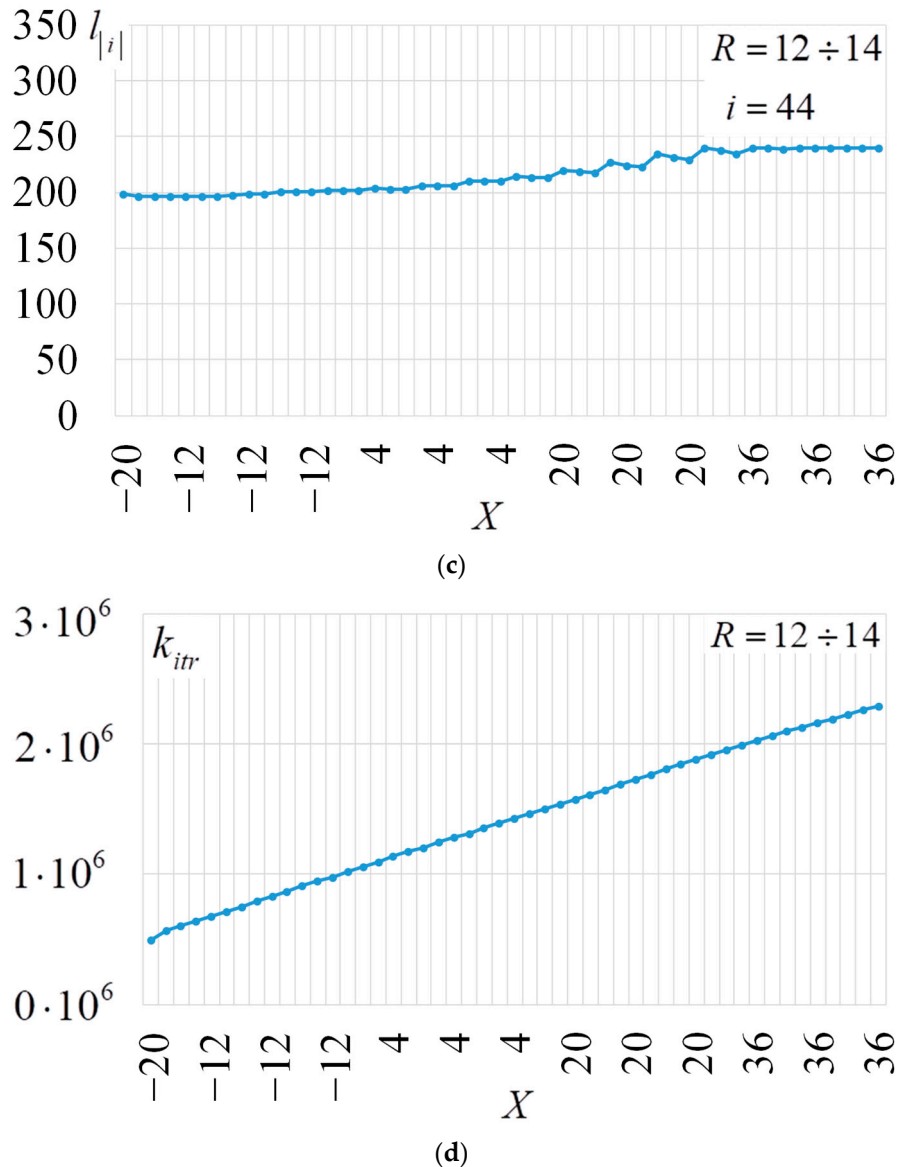

**Figure 22.** Trajectory of the $l_{|i|}$ S-box at dynamical change of parameters for $R = 12 \div 14$.

Despite the fact that the total number of iterations increases with an increasing $R$ (Figures 17–22d), the best result is shown by cycles with a smaller value of $R$. For all cycles, the value $l_{|i=52|} = 0$ is achieved (Figures 17–22a).

The dynamics of achieving the minimum value $l_{|i|}$ are summarized in Table 5.

**Table 5.** Values of parameters $R$ and $X$ at which the minimum value $l_{|i|}$ is achieved.

| | i = 52 | |
|---|---|---|
| R | X | l\|i\| |
| $R \in [2, 3, 4]$ | $X \geq 44$ | $l_{|i|} = 0$ |
| $R \in [4, 5, 6]$ | $X \geq 32$ | $l_{|i|} = 0$ |
| $R \in [6, 7, 8]$ | $X \geq 36$ | $l_{|i|} = 0$ |
| $R \in [8, 9, 10]$ | $X \geq 28$ | $l_{|i|} = 0$ |
| $R \in [10, 11, 12]$ | $X \geq 24$ | $l_{|i|} = 0$ |
| $R \in [12, 13, 14]$ | $X \geq 20$ | $l_{|i|} = 0$ |

**Table 5.** *Cont.*

| i = 48 | | |
|---|---|---|
| **R** | **X** | **l\|i\|** |
| R ∈ [2, 3, 4] | X ≥ 44 | l\|i\| = 21 |
| R ∈ [4, 5, 6] | X ≥ 36 | l\|i\| = 20 |
| R ∈ [6, 7, 8] | X ≥ 36 | l\|i\| = 21 |
| R ∈ [8, 9, 10] | X ≥ 36 | l\|i\| = 22 |
| R ∈ [10, 11, 12] | X ≥ 36 | l\|i\| = 25 |
| R ∈ [12, 13, 14] | X ≥ 36 | l\|i\| = 28 |
| **i = 44** | | |
| **R** | **X** | **l\|i\|** |
| R ∈ [2, 3, 4] | 12 ≥ X ≥ 28 | l\|i\| = 150 |
| R ∈ [4, 5, 6] | X ≤ 20 | l\|i\| = 160 |
| R ∈ [6, 7, 8] | X ≤ 20 | l\|i\| = 160 |
| R ∈ [8, 9, 10] | X ≤ 4 | l\|i\| = 160 |
| R ∈ [10, 11, 12] | X ≤ 4 | l\|i\| = 180 |
| R ∈ [12, 13, 14] | X ≤ 4 | l\|i\| = 200 |

The use of a small change in the parameter *R* (up to three units) allowed us to reduce the average statistical value $l_{|i=48|}$ by another two (to $l_{|i=48|}$ = 20) compared to the best result obtained in the first series of tests.

5.5.3. Pushing the Boundaries: Nonlinearity and Computational Efficiency

In the third series of experiments, we push the boundaries of our algorithm by exploring its performance under challenging scenarios, such as generating S-boxes with extremely high nonlinearity. Through comprehensive visualizations and detailed explanations, we showcase the limitations and potential of our approach in pushing the state-of-the-art in S-box generation.

These tests were conducted with a linear increase in the parameter *R* and a proportional increase in the parameter *X*. The results were averaged over 100 runs. In each run, a cycle of parameter change was performed and the following chains were followed (see Table 6).

**Table 6.** Selected parameters *X* and *R* in the third series of experiments.

| X = −36 + 4·R | | | | | | | | | | | | | | |
|---|---|---|---|---|---|---|---|---|---|---|---|---|---|---|
| R | 5 | 6 | 7 | 8 | 9 | 10 | 11 | 12 | 13 | 14 | | | | |
| X | −16 | −12 | −8 | −4 | 0 | 4 | 8 | 12 | 16 | 20 | | | | |
| **X = −28 + 4·R** | | | | | | | | | | | | | | |
| R | 4 | 5 | 6 | 7 | 8 | 9 | 10 | 11 | 121 | 3 | 14 | 15 | 16 | 17 | 18 |
| X | −12 | −8 | −4 | 0 | 4 | 8 | 12 | 16 | 456 | −16 | 28 | 32 | 36 | 40 | 44 |

The results are shown in Figures 23 and 24 (the values of the change in *R* are not shown in the figure).

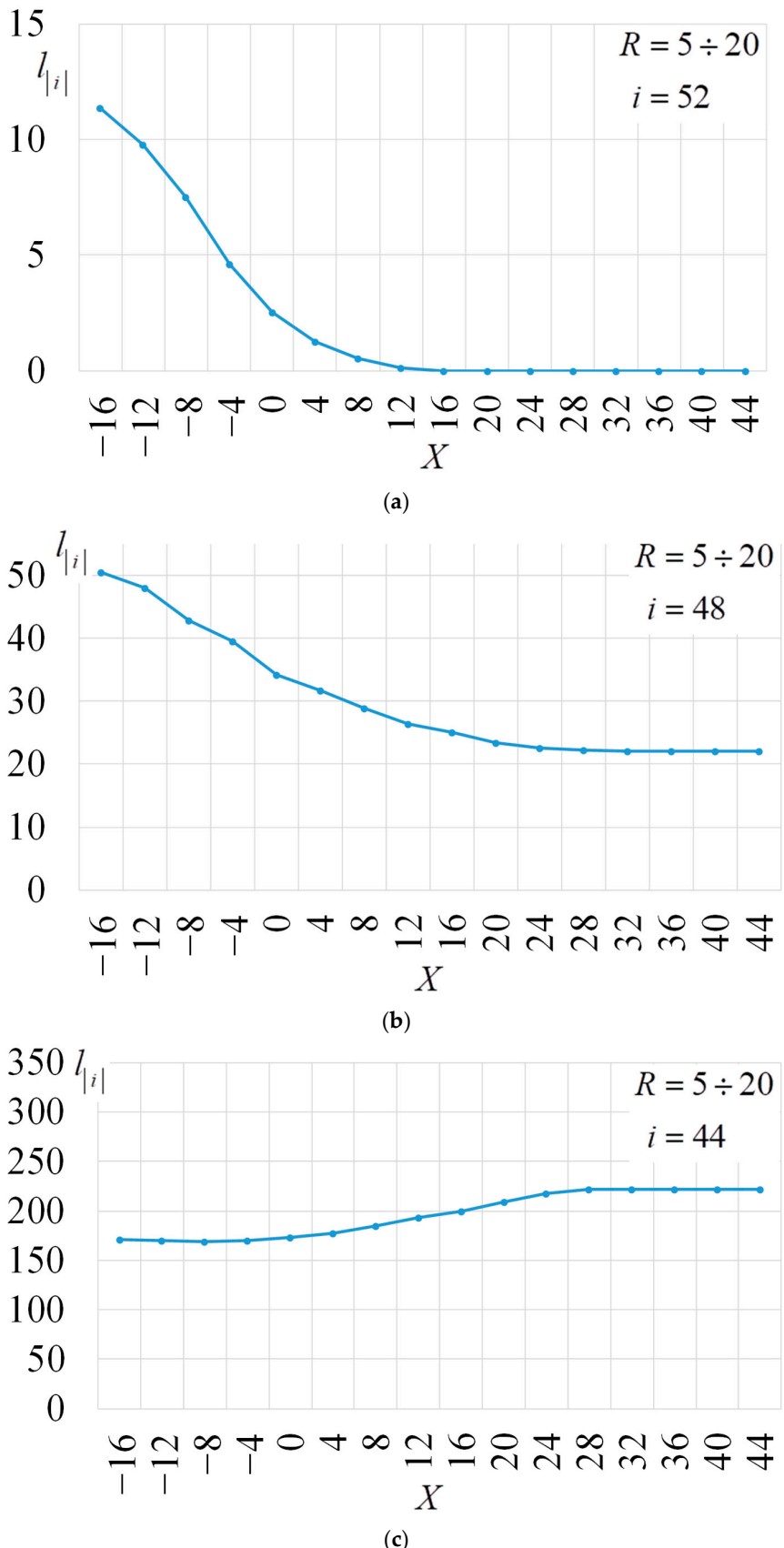

**Figure 23.** *Cont.*

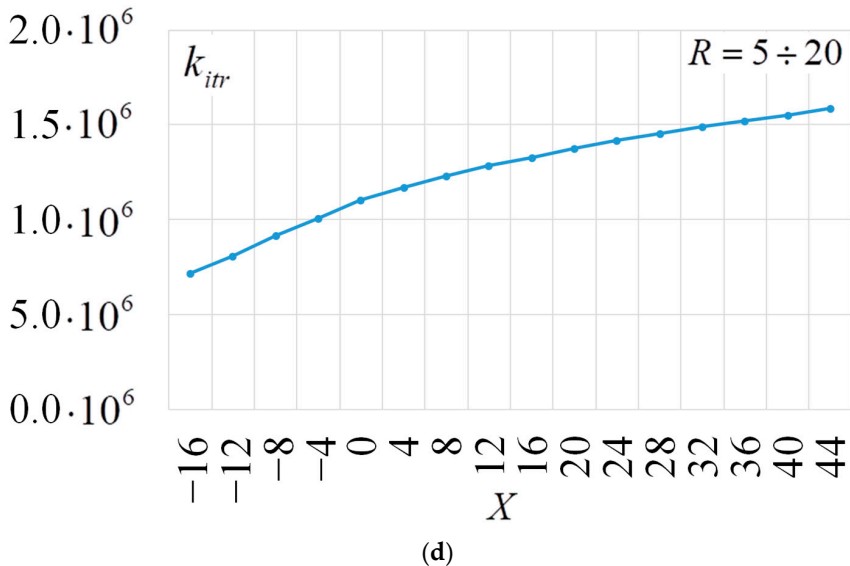

**(d)**

**Figure 23.** Trajectory of the $l_{|i|}$ S-box at dynamical change of parameters for $R = 5 \div 20$.

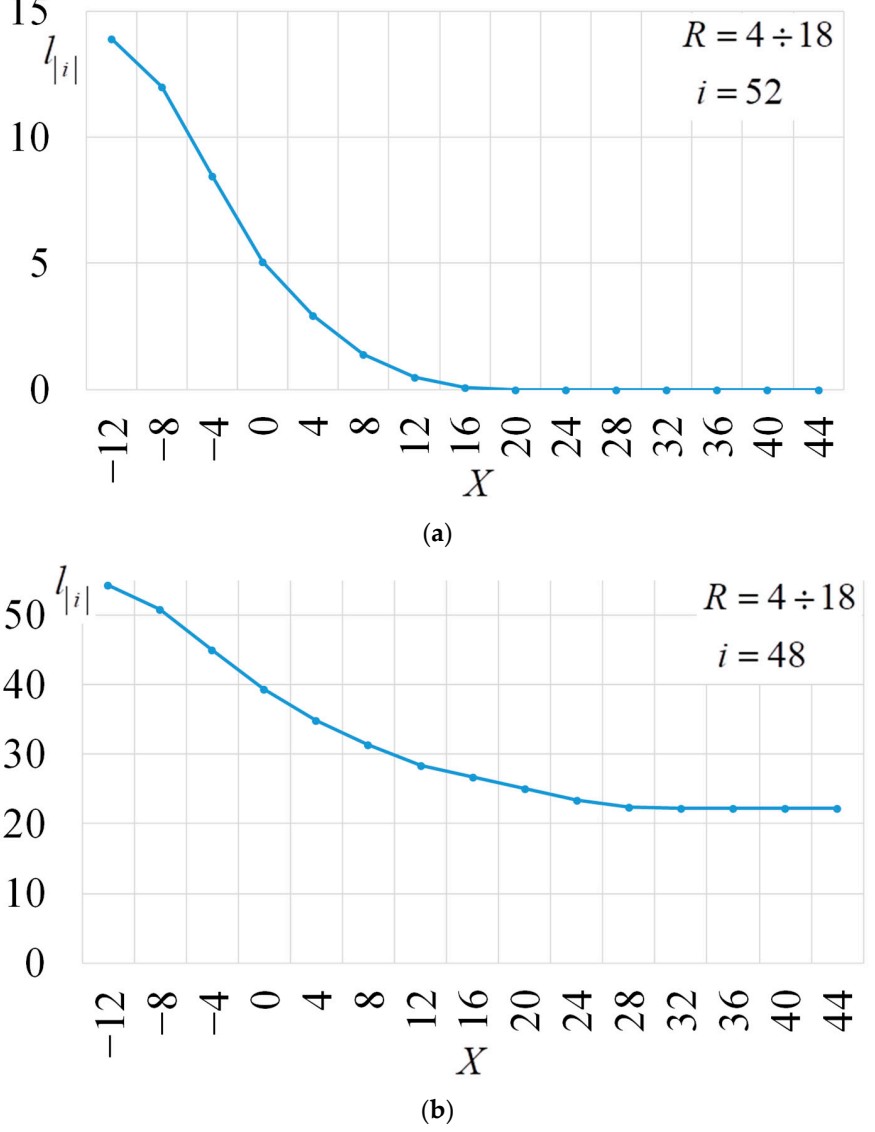

**(a)**

**(b)**

**Figure 24.** *Cont.*

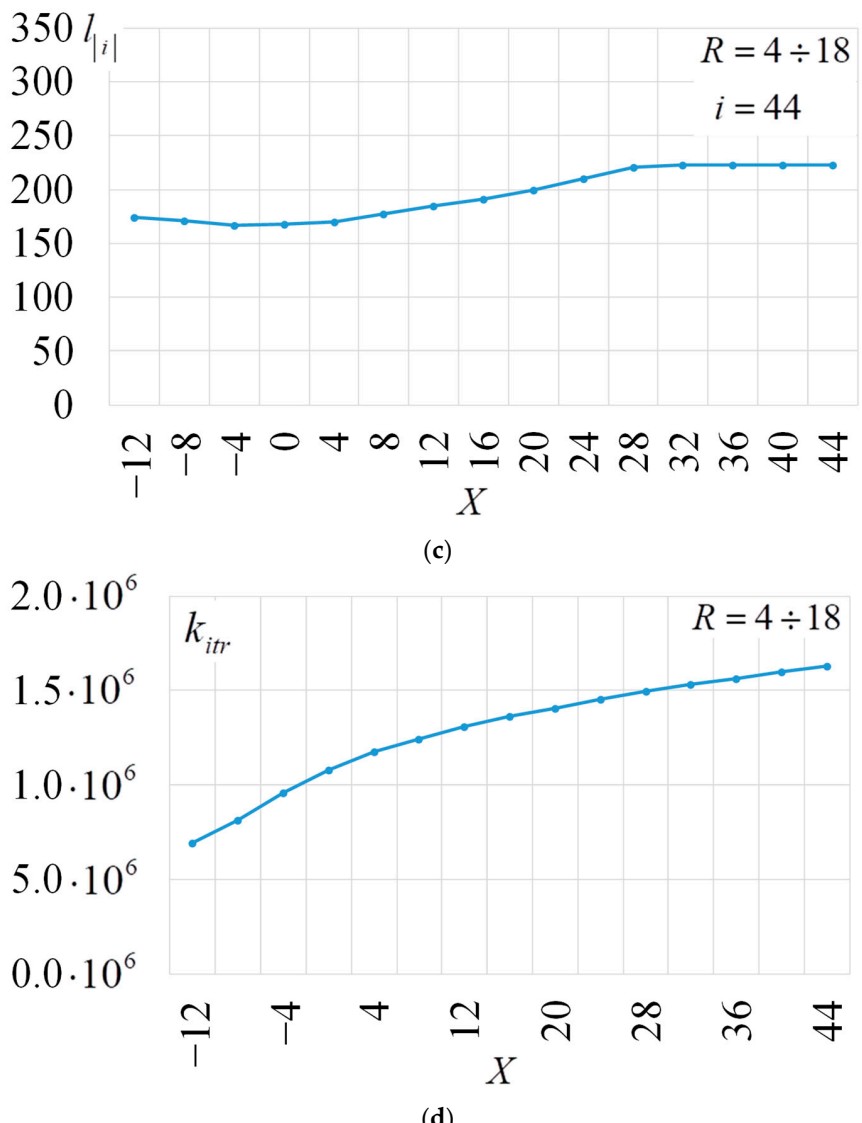

**Figure 24.** Trajectory of the $l_{|i|}$ S-box at dynamical change of parameters for $R = 4 \div 18$.

With the above parameters, no improvement was achieved in the third series. The average statistical value $l_{|i=48|}$ for both cycles was about $l_{|i=48|} = 22$). The average number of iterations of the search algorithm was about $1.5 \times 10^6$ iterations.

## 6. Discussions

Our study delved deep into the heuristic methods of nonlinear substitution search, commonly known as S-boxes. This pivotal research area ties directly to enhancing the robustness of cryptographic algorithms that leverage a secret key. By injecting nonlinearity through these S-boxes, resilience against cryptanalytic attacks receives a significant boost.

We embarked on this journey with a very specific aim: the generation of 8-bit bijective substitutions with a nonlinearity benchmarked at 104 or higher. Our findings revealed that the traditional static methodologies have intrinsic limitations. This realization led us to the conceptualization of a novel approach to S-box generation—dynamic heuristic methods with evolving parameters for the cost function.

### 6.1. Comparison with State of the Art

The narrative of research, especially within the realm of heuristic search and cryptography, is enriched when engaged in comparative analysis. Table 7 delineates a discerning

landscape where diverse methodologies are juxtaposed, unraveling the intricacies and performance metrics of various search methods in generating S-boxes. Herein, our exploration unfolds along two pivotal dimensions: the 'Average number of iterations' and the 'Frequency of achieving the target outcome'. A closer, nuanced examination of these indices elucidates the layered complexities and insights nestled within these numbers.

**Table 7.** Results of a comparative analysis of known techniques for generating S-boxes with nonlinearity 104.

| Literary Source | Search Method | Average Number of Iterations | Frequency of Achieving the Target Outcome |
|:---:|:---:|:---:|:---:|
| [44] | Simulated Annealing | over 3,000,000 | 1/200 (for generating S-boxes with nonlinearity 102) |
| [16] | Genetic and Tree | over 3,000,000 | no data |
| [6] | Simulated Annealing | around 3,000,000 | no data |
| [45] | Simulated Annealing | around 450,000 | 56.4% |
| [19] | Genetic and Tree | over 160,000 | no data |
| [17] | Hill climbing | over 70,000 | no data |
| [20] | Hill climbing | over 65,000 | 11/30 |
| **Our work** | **Hill climbing, dynamic selection of cost function parameters** | **around 50,000** | **100%** |

- Average Number of Iterations: This metric manifests as a tangible, quantitative indicator of computational effort, encapsulating the average exertion required by an algorithm to gravitate towards a solution. In heuristic landscapes, this becomes particularly poignant, embodying the exploration–exploitation balance and algorithmic efficiency.
- Frequency of Achieving the Target Outcome: This transcends mere numeric representation, dovetailing into the stochastic nature of heuristic methods. Recognizing that achieving the target is a probabilistic event, this metric epitomizes an algorithm's reliability and robustness in diverse scenarios and over multiple runs.

Pitting diverse methodologies against one another within a unified framework epitomizes a subtle art, ensconcing the principle of universal comparability. While classical approaches tether themselves to determinism and rigidity, our comparative matrix here invokes a spectrum of heuristic methodologies and objectively scrutinizes them under a universally applicable lens. The essence of this methodology pivots on its capability to furnish a coherent, relatable comparison amidst the probabilistic and non-deterministic nature of heuristic algorithms.

The comparative analysis presented in Table 7 reveals several key insights into the performance of various heuristic search techniques for S-box generation.

Firstly, it is evident that early approaches, such as Simulated Annealing [6,13] and Genetic Algorithms [14–16], suffer from high computational costs. These methods often require millions of iterations to find S-boxes with the desired nonlinearity. Moreover, their success rates are either low or not reported, indicating a lack of consistency and reliability. For instance, the Simulated Annealing approach in [6,44] achieved a success rate of only 1/200 for finding S-boxes with a nonlinearity of 102, while requiring over 3,000,000 iterations.

However, more recent studies have demonstrated a clear trend of improvement. The works of [7,19,20] showcased significant reductions in the number of iterations needed, as well as higher success rates. This progress can be attributed to advancements in heuristic techniques, such as the development of more efficient cost functions and the fine-tuning of algorithm parameters. For example, the hill-climbing approach employed by Freyre-Echevarría et al. [17,20], which introduces the WCFS cost function, requires an average of 65,000 iterations to find S-boxes with a nonlinearity of 104, with a success rate of 36.7%.

Our work builds upon these advancements and achieves a substantial leap forward in both computational efficiency and reliability. By integrating dynamic parameter adjustment into the hill-climbing algorithm, we reduce the average number of iterations to just 50,000—a significant improvement over the previous state of the art. Furthermore, our approach guarantees a 100% success rate in finding S-boxes with a nonlinearity of 104 or higher, demonstrating a level of consistency and robustness that is unmatched by prior methods.

### 6.2. Significance of Achieved Results

The implications of our findings are far-reaching. In the context of cryptographic engineering, where the efficiency and reliability of S-box generation are of paramount importance, our approach offers a promising solution. By dramatically reducing the computational cost and ensuring a high success rate, our method paves the way for more efficient and secure cryptographic systems. Moreover, the dynamic parameter adjustment technique introduced in our work has the potential to be applied to other optimization problems in cryptography and beyond, opening up new avenues for research and innovation.

To gain deeper insights, let us dissect the hill-climbing algorithm employed in our research. Our version of this algorithm integrated the WCFS heuristic cost function from reference [20]. The focus was unambiguous: to ascertain the potential of dynamic parameter adjustments in amplifying the algorithm's efficiency. The graphical data from our study affirm our hypothesis. The declining trend in the maximum absolute values of the Walsh–Hadamard coefficients illustrates that such parameter modifications can indeed expedite the generation process. In simpler terms, higher nonlinearity among the S-box alternatives in the search space can be achieved in fewer iterations.

While our results were promising, they were not without challenges. Generating a random 8-bit bijective S-box with a nonlinearity of 106 remained elusive. Nevertheless, we came tantalizingly close. The histogram of Walsh–Hadamard coefficients (as depicted in Figure 25) for the most optimal random S-box we unearthed speaks volumes. With just 12 coefficients standing tall with $|WHT[b,i]| > 44$, the implications are clear: only these 12 values need transformation to reach the coveted:

$$N(S)\frac{256 - \max_{b,i}|WHT(b,i)|}{2} = 106.$$

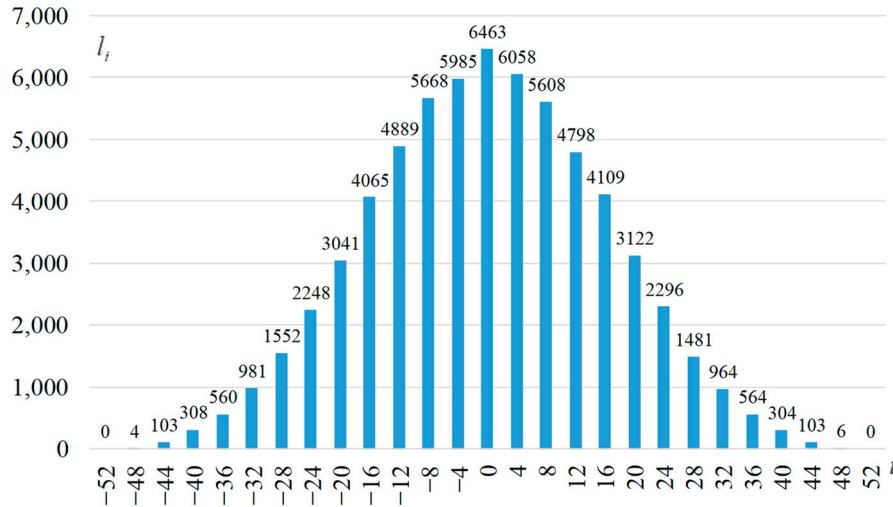

**Figure 25.** Histogram of Walsh–Hadamard coefficients for the best found random S-box.

In the grand tapestry of research in this domain, our findings are notable. To our understanding, the results we achieved are unmatched in the current literature, spotlighting the potential of our approach. The gaps in the current research field, particularly

around efficient methods for generating S-boxes, are being progressively filled by dynamic methodologies like ours. The uncharted territory of obtaining random S-boxes with $N(S) = 106$ remains, and it is this tantalizing challenge that forms the crux of our future research endeavors.

In wrapping up, our research serves as a pivotal point, introducing an innovative dynamic methodology to the S-box generation discourse. It bears a conceptual kinship with dynamic programming—a computer science paradigm predicated on breaking down larger problems into smaller, more manageable subproblems. Dynamic programming, renowned for its optimization procedures, often capitalizes on previously solved subproblems, negating the need for redundant recalculations. Drawing parallels, our approach, by dynamically adjusting cost function parameters, also inherently leverages past learnings to better inform the ongoing search process.

*6.3. Limitations of the Results*

While our method has made substantial progress towards generating S-boxes with optimal cryptographic properties, it is important to acknowledge the limitations and challenges that remain. Achieving a nonlinearity of 106 for 8-bit bijective S-boxes is still an open problem, and our approach, while coming closer than many previous methods, has not yet reached this goal. Nevertheless, the quantitative results presented in this study demonstrate the potential for further improvements and serve as a benchmark for future research.

## 7. Conclusions

This study explores the application of heuristic methods for constructing nonlinear substitution boxes (S-boxes), which are crucial components in symmetric key cryptography. Our research focuses on generating 8-bit bijective S-boxes with high nonlinearity, specifically targeting values of 104 and above.

The key innovation of our approach lies in the dynamic adjustment of cost function parameters within the heuristic search process. This adaptive mechanism allows for a more efficient exploration of the vast search space, leading to improved results compared to traditional static methods. The comparative analysis in Table 7 quantitatively demonstrates the superiority of our approach in terms of both iteration count and success rate.

On average, our method requires only 50,000 iterations to find an S-box with the desired nonlinearity, a significant reduction compared to the 65,000 iterations reported by Freyre-Echevarría et al. [17,20] and the 450,000 iterations needed in [45]. Moreover, our approach guarantees a 100% success rate, a substantial improvement over the 36.7% achieved by Freyre-Echevarría et al. [17,20] and the 56.4% reported in [45]. These quantitative metrics highlight the efficiency and reliability of our method.

The effectiveness of our approach is further evidenced by the reduction in Walsh–Hadamard coefficient values, as illustrated in Figure 25. By dynamically adjusting the cost function parameters, we are able to minimize the number of coefficients with high absolute values, thus improving the nonlinearity of the generated S-boxes. Specifically, our method consistently produces S-boxes with only 12 coefficients exceeding the absolute value of 44, a significant achievement in the context of 8-bit bijective S-boxes.

From a broader perspective, our work establishes a quantitative foundation for the integration of dynamic programming principles into the field of cryptographic transformation. The dynamic parameter adjustment technique introduced in this study showed promising results in terms of efficiency and reliability, and its potential applications extend beyond S-box generation to other optimization problems in cryptography. Recent studies in fractional calculus, such as the analysis of Kuramoto–Sivashinsky equations with non-singular kernel operators [46], the investigation of fractional-order Helmholtz equations in two space dimensions [47], and the exploration of symmetric soliton solutions for the fractional coupled Konno–Onno system [48], highlight the potential for interdisciplinary collaborations in applied mathematics and computational physics. These topics inspire us

to consider the broader implications and future directions of our work in the context of advancing scientific knowledge across various domains. Furthermore, as supported by recent studies [49,50], the resistance of S-boxes to side-channel attacks, particularly power analysis attacks, is a critical aspect that requires further investigation. Future research should focus on developing heuristic methods that not only optimize the cryptographic properties of S-boxes, but also incorporate techniques to enhance their resistance to side-channel attacks. This direction is crucial to ensure the long-term security of symmetric key cryptographic algorithms, even in the face of evolving threats such as post-quantum attacks.

**Author Contributions:** Conceptualization, O.K.; methodology, O.K.; resources, N.P.; formal analysis, M.K. and E.F.; investigation, E.F. and R.S.; software and validation, S.K.; writing—original draft preparation, O.K., M.K. and R.S.; writing—review and editing, O.K., M.K. and R.S.; funding acquisition, R.S. All authors have read and agreed to the published version of the manuscript.

**Funding:** This project has received funding from the European Union's Horizon 2020 research and innovation programme under the Marie Skłodowska-Curie grant agreement No. 101007820—TRUST. This publication reflects only the author's view and the REA is not responsible for any use that may be made of the information it contains. This research was funded by the European Union—NextGenerationEU under the Italian Ministry of University and Research (MIUR), National Innovation Ecosystem grant ECS00000041-VITALITY-CUP D83C22000710005.

**Data Availability Statement:** The datasets generated during and/or analyzed during the current study are available from the corresponding author on reasonable request.

**Conflicts of Interest:** The authors declare no conflicts of interest.

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
