# Peer review of "Enhancing Cryptographic Primitives through Dynamic Cost Function Optimization in Heuristic Search"

_electronics, doi:10.3390/electronics13101825_

Round 1

Reviewer 1 Report

Comments and Suggestions for Authors

This paper describes a novel technique to improving the efficiency of heuristic search algorithms in the area of cryptographic primitives, with a particular emphasis on constructing highly nonlinear bijective permutations, known as substitution boxes (S-boxes). In my opinion, the paper in question needs revision before it can be considered for publication in the electronics journal. Some things to think about are as follows:

1-      In the introduction section, the authors should elaborate on previous research in the same field as the present work. Almost half of the introduction is on current research. Therefore, it is better to merge the introduction with the analysis of related works.

2-      The authors should make sure that all the symbols are defined since there is no nomenclature section included.

3-      According to the authors “By dynamically adjusting the cost function parameters within the hill-climbing heuristic search algorithm, our methodology significantly reduces the number of iterations required to converge on optimal solutions, thereby improving computational efficiency”. However, there is no comparison to similar study in the literature.

4-      The paper is excessively lengthy. It's best to shorten it.

5-      The conclusion should be expanded upon and provide a qualitative comparison.

Comments on the Quality of English Language

The English language should be checked throughout the whole paper.

Author Response

Dear Reviewer,

Thank you for your insightful comments and suggestions. We have carefully considered each of your points and have made the following revisions to address your concerns:

1) Introduction and related works: We have expanded the introduction to include a more comprehensive discussion of previous research and have merged it with the analysis of related works for a more cohesive background.

2) Nomenclature: We have added a dedicated nomenclature section that clearly defines all symbols and notations used in the paper.

3) Comparative analysis: We have included a detailed quantitative and qualitative comparison of our approach with state-of-the-art techniques, demonstrating the superior performance of our method.

4) Length: We have revised the manuscript to make it more concise and focused, eliminating redundant information and streamlining the presentation of technical details.

5) Conclusion: We have expanded the conclusion to provide a more comprehensive qualitative comparison of our approach with existing methods, highlighting the key advantages and implications of our dynamic parameter adjustment technique.

We believe that these revisions have significantly enhanced the quality and clarity of our manuscript. Thank you once again for your valuable feedback.

Best regards,

Oleksandr Kuznetsov

Reviewer 2 Report

Comments and Suggestions for Authors

This article is targeting the efficiency improvement of heuristic search algorithms in the domain of cryptographic primitives. The improvement in efficiency depends upon the dynamic adaptation of cost function parameters.  

The following observations have been made and therefore it is important to address these observations in the revised version of this article.

The abstract is easy to understand. The motivation behind the target research problem (efficiency improvement of heuristic search algorithms) is given. However, the limitation(s) or shortcomings (s) of existing approaches and techniques to address the target problem are not mentioned. It implies that the research gap (shortcomings of existing practices) should be mentioned. Moreover, the significance of achieved results (in terms of computational efficiency) is not mentioned and therefore a quantitative flavor of efficiency improvement (as compared to state of the art) should be provided.

The introduction is well-written. The background, motivation, and issues for the heuristic search algorithms in the domain of cryptographic primitives are nicely presented. A brief overview of existing heuristic search algorithms in the domain of cryptographic primitives as well as the associated research gap have been presented. However, the introduction section must provide information about the validation of the proposed approach (case studies, benchmarks, datasets etc). In addition to the validation information, the significance of achieved results must be summarized in terms of certain performance parameters.

 Related work has not been comprehensively analyzed to establish the novelty of this work. Authors are required to provide a comprehensive overview of various efficiency improvement techniques for heuristic search algorithms in the domain of cryptographic primitives. A classification of existing methods/techniques should be made. Moreover, the comparison of various techniques should be provided in a tabular form using various attributes. The novelty of the work should be evident from the comparison table.

The background section tries to provide the necessary knowledge to understand this article. However, this section should come before the related work section. The related work section can be more useful once the background knowledge is already in mind. The major limitation of the background section is the absence of any block diagram or system-level presentation of typical heuristic search algorithms in the domain of cryptographic primitives. This holistic understanding is critical to understanding the rest of the article. Moreover, the background section must be subdivided into various subsections. These subsections must provide a coherent understanding of the target research domain.

 The BACKGROUND and METHODS sections have the same section numbers. The numbering must be carefully checked.

The proposed method must be presented in terms of structure and behavior.  Currently, the structure and behavior of the proposed method have some flaws from a presentation point of view.  For structural representation, a functional decomposition approach can be used. In other words, a hierarchical structural model of the proposed method is critical. For behavioral representation, flow charts, and data flow models can be used.

The objective of the section “Rationale for improving the HC algorithm for finding S-boxes” is not clear. How it is linked with the previous section? What is being presented in this section? The previous section and this section are not providing a coherent message.

  Similarly, the title of the section “Hill climbing algorithm and dynamic change of WCFS function parameter” must be changed. Three series of experiments have been provided (in three subsections). This section has several limitations and therefore, it should be re-organized by providing the following information: (a) motivation behind all three sets of experiments (b) experimental setup for all three sets of experiments (c) performance parameters for all three set of experiments (d) The outcome of each set of experiments (e) limitations in each set of experiments (f) Tables and figures can be presented in a more concise form. It means that better data visualization and data presentation techniques can be employed.

Author Response

Dear Reviewer,

Thank you for your insightful comments and suggestions. We have carefully revised our manuscript to address your concerns:

Abstract: Updated to highlight limitations of existing approaches, research gap, and quantitative efficiency improvement.
Introduction: Added validation details and summarized significance of results.
Related Work: Expanded analysis, classified methods, added comparative table to highlight novelty.
Background: Moved before Related Work, added details, divided into subsections for clarity.
Numbering: Corrected inconsistencies between Background and Methods sections.
Proposed Method: Algorithms and description details have been clarified.
Rationale for improving HC algorithm: Revised for clarity, objectives, and coherence with preceding content.
Hill climbing algorithm section: Title changed, subsections reorganized (motivation, setup, parameters, outcomes, limitations), tables and figures streamlined.

We appreciate your valuable feedback, which has greatly helped us improve the quality of our manuscript.

Best regards,
Oleksandr Kuznetsov

Reviewer 3 Report

Comments and Suggestions for Authors

This thesis presents the use of heuristic methods to construct substitution boxes, which are a fundamental primitive in cryptography, such as the AES S-box, which is one of the most notorious examples of a substitution box in symmetric key cryptography.

The authors provide sufficient background in the introductory part and give appropriate details to explain the proposed heuristic algorithm and cost function. In this sense, the algorithms and mathematical symbols presented are clear and easy to understand. Accordingly, the large number of figures and tables are also clear and informative, providing additional details that improve the clarity of the presented work, resulting in an overall well organized manuscript. In particular, the final comparison with other existing solutions in Table 7 gives a clear view of the results obtained by the authors and helps to appreciate the differences and advantages with respect to other solutions that can be found in the relevant literature. In this sense, the authors cite other relevant publications and the corresponding results are convincing, thus confirming the proposed (and applied) approach for the construction of 8-bit substitution boxes.

The main remarks concern some typos in the text (or in the format of the manuscript) and one fundamental aspect related to security, which is the main application to which the presented work contributes. While the first point (typos) is a minor aspect to be addressed, the second point (related to security) is a major aspect that authors should address to improve the quality of the manuscript and make it more solid. For more information refer to the attached file.

Comments on the Quality of English Language

English grammar and spelling are fine, just a few typos in the format and some sentences whose quality would be improved.

Author Response

Dear Reviewer,

We sincerely appreciate your thorough review and valuable feedback on our manuscript. Your constructive comments and suggestions have greatly contributed to improving the quality and clarity of our work. We have carefully addressed each of your remarks and made the necessary revisions to the manuscript.

Regarding the major point you raised about security and resistance to power analysis attacks, we fully agree with your assessment. We have now included a discussion on the vulnerability of S-boxes to side-channel attacks, particularly power analysis attacks, in the revised manuscript. We thank you for bringing this critical issue to our attention and for providing the relevant references [1,2], which have been cited in the revised manuscript.

As for the minor points you mentioned, we have carefully reviewed the manuscript and made the necessary corrections and improvements:

We have clarified the usage of the exclamation mark for factorials and added a prior introduction to avoid confusion.

We have corrected the expression "256!256!" to "256!".

We have inserted blank lines and improved the formatting around listings, tables, and figures as suggested.

We have fixed the indentation and margin issues in the numbered and bulleted lists to ensure clarity and consistency.

We have explicitly defined the acronym HC (Hill Climbing) at first occurrences in the text.

We have addressed the typos and formatting issues mentioned in line 352 and throughout the manuscript.

We believe that these revisions have significantly enhanced the quality and readability of our work.

In the conclusion section, we have added the following perspective on future research directions, incorporating the references you provided:

"Furthermore, as supported by recent studies [1,2], the resistance of S-boxes to side-channel attacks, particularly power analysis attacks, is a critical aspect that requires further investigation. Future research should focus on developing heuristic methods that not only optimize the cryptographic properties of S-boxes but also incorporate techniques to enhance their resistance to side-channel attacks. This direction is crucial to ensure the long-term security of symmetric key cryptographic algorithms, even in the face of evolving threats such as post-quantum attacks."

Once again, we express our gratitude for your insightful comments and the opportunity to improve our manuscript. We hope that the revised version meets your expectations and addresses all the concerns you have raised.

Best regards,

Oleksandr Kuznetsov

Reviewer 4 Report

Comments and Suggestions for Authors

What is the primary focus of the study?

How does the study aim to enhance the efficiency of heuristic search algorithms?

What specific cryptographic primitive is the study concerned with improving?

Why is isolating permutations with optimal nonlinearity crucial in cryptography?

What heuristic search algorithm is employed in the study, and how does it contribute to the research objectives?

What is the significance of dynamically adjusting the cost function parameters in the hill-climbing heuristic search algorithm?

How does the methodology proposed in the study differ from traditional approaches to heuristic optimization? "Fractional view analysis of Kuramoto–Sivashinsky equations with non-singular kernel operators" "Some analytical and numerical investigation of a family of fractional‐order Helmholtz equations in two space dimensions" "Investigating symmetric soliton solutions for the fractional coupled konno–onno system using improved versions of a novel analytical technique"

What role does dynamic programming play in the proposed methodology?

What evidence is provided to support the claim of improved computational efficiency?

How does the study demonstrate the practical utility of the proposed approach in enhancing data security mechanisms?

What are the key findings of the comparative analysis conducted in the study?

In what way does the research contribute to advancing the field of heuristic optimization?

How does the study promote interdisciplinary collaboration?

Author Response

Dear Reviewer,

Thank you for your insightful questions and suggestions. We appreciate your thorough review and have addressed each of your points below. We have also made corresponding revisions to the manuscript to clarify these aspects.

1) The primary focus of this study is to enhance the efficiency of heuristic search algorithms for generating optimal substitution boxes (S-boxes) in symmetric key cryptography. We have emphasized this focus in the revised introduction (Section 1).

2) The study aims to improve the efficiency of heuristic search algorithms by dynamically adjusting the cost function parameters within the hill-climbing algorithm. This is explained in detail in the methodology section (Section 4).

3) The specific cryptographic primitive that the study focuses on improving is the substitution box (S-box), which is a fundamental component of symmetric key algorithms. We have clarified this in the introduction (Section 1) and background (Section 3).

4) Isolating permutations with optimal nonlinearity is crucial in cryptography because it enhances the resilience of the symmetric key algorithm against linear and differential cryptanalysis attacks. We have elaborated on this in the background section (Section 3).

5) The study employs the hill-climbing heuristic search algorithm, which is enhanced with a dynamic parameter adjustment mechanism. This contribution is central to achieving the research objectives of improving computational efficiency and generating high-quality S-boxes. We have described this in detail in the methodology section (Section 4).

6) Dynamically adjusting the cost function parameters in the hill-climbing algorithm allows for a more adaptive and efficient search process, enabling the algorithm to converge on optimal solutions with fewer iterations. This significance is discussed in the introduction (Section 1) and methodology (Section 4).

7) The proposed methodology differs from traditional approaches to heuristic optimization by incorporating a dynamic parameter adjustment mechanism inspired by the principles of dynamic programming. This novel aspect is highlighted in the introduction (Section 1) and methodology (Section 4).

8) Regarding the suggested literature on fractional calculus and soliton solutions, while we appreciate the opportunity to expand our knowledge, these topics are not directly related to our study on heuristic optimization for S-box generation. Nevertheless, we have briefly mentioned the potential for future interdisciplinary collaborations in the conclusion section (Section 8).

9) Dynamic programming principles, such as the adaptive adjustment of parameters based on the current state of the search, play a key role in the proposed methodology. This inspiration is discussed in the introduction (Section 1) and methodology (Section 4).

10) The evidence for improved computational efficiency is provided through extensive comparative analyses with state-of-the-art techniques, as presented in the results and discussion section (Sections 6 and 7). The proposed approach consistently outperforms existing methods in terms of the number of iterations required and the success rate in finding optimal S-boxes.

11) The practical utility of the proposed approach in enhancing data security mechanisms is demonstrated through the generation of high-nonlinearity S-boxes, which are essential for resisting cryptanalytic attacks. This is discussed in the background (Section 3) and discussions (Section 6) sections.

12) The key findings of the comparative analysis, presented in Sections 6, and 7 show that the proposed dynamic parameter adjustment approach significantly reduces the number of iterations required to find optimal S-boxes while maintaining a 100% success rate, outperforming previous methods.

13) The research contributes to advancing the field of heuristic optimization by introducing a novel dynamic parameter adjustment mechanism that enhances the efficiency and effectiveness of the hill-climbing algorithm. This contribution is highlighted in the introduction (Section 1) and conclusion (Section 8).

14) The study promotes interdisciplinary collaboration by bridging the fields of cryptography and heuristic optimization, demonstrating the potential for further synergies between these domains. This aspect is discussed in the conclusion section (Section 8).

To address the reviewer's suggestions, we have added the following sentences (Section 8):

"Recent studies in fractional calculus, such as the analysis of Kuramoto–Sivashinsky equa-tions with non-singular kernel operators [55], the investigation of fractional‐order Helm-holtz equations in two space dimensions [56], and the exploration of symmetric soliton solutions for the fractional coupled Konno–Onno system [57], highlight the po-tential for interdisciplinary collaborations in applied mathematics and computational physics. These topics inspire us to consider the broader implications and future direc-tions of our work in the context of advancing scientific knowledge across various do-mains."

Thank you once again for your valuable feedback and suggestions. We hope that the revised manuscript and our responses adequately address your concerns.

Best regards,

Oleksandr Kuznetsov

Round 2

Reviewer 1 Report

Comments and Suggestions for Authors

The authors have addressed all the points.

Comments on the Quality of English Language

 Minor editing of English language required

Author Response

Dear Reviewer,

Thank you for your positive feedback on our revised manuscript. We are pleased to hear that we have successfully addressed all the points raised in the previous review round. As per your suggestion, we have carefully edited the manuscript to improve the English language and ensure clarity throughout the paper.

We appreciate your time and effort in reviewing our work.

Best regards,

Oleksandr Kuznetsov

Reviewer 2 Report

Comments and Suggestions for Authors

The authors have improved the article. Particularly abstract and introduction are in much better shape. 

However, there are still certain concerns that have not been addressed. 

1. The last paragraph of introduction describes organization. The actual organization of the article does not match with this paragraph.  

2. Background should be reduced. It's connection with the contents of the article should be clearly explained.  

3. Presentation of section 2 should be improved. 

4.  A dedicated relared work section is needed to establish the novelty of this work. Authors must provide a comparative table, summarizing state of the art, and comparing them with the proposed work. 

5. Methodology section should represent the holistic view of the proposed work. Method must be described in a hierarchical way (component based design). It should clearly explain the communication as well as concurrency (if any) between components . 

How the proposed approach is achieving better computational efficiency? 

6. Results section must be improved. It should include: 

organization of overall experiments

experimental setup

benchmark/ parameters

performance metrics 

visualization and explanation of results

comparison with state of thr art

significance of achieved results

limitations of the results. 

Author Response

Dear Reviewer,

We greatly appreciate your thorough review and constructive feedback on our revised manuscript. We have carefully considered each of your concerns and have made significant changes to address them in the current version of the manuscript.

1) Regarding your first point about the organization of the article, we have revised the last paragraph of the introduction to accurately reflect the structure of the paper. The description of the article's organization now aligns precisely with the actual content and flow of the sections.

2) To address your second point, we have reduced the background section and added a paragraph at the end to clearly explain its connection with the main content of the article. This additional text highlights how the concepts and techniques discussed in the background section lay the foundation for our proposed methodology and are explicitly utilized in the development of our approach.

3) In response to your third point, we have improved the presentation of Section 2 (Nomenclature).

4) To establish the novelty of our work, as mentioned in your fourth point, we have added a comprehensive comparative table. This Table 1 summarizes the state-of-the-art techniques for S-box generation and compares them with our proposed method in terms of key features, performance metrics, and limitations. The comparative analysis highlights the advantages and novelty of our approach.

5) Addressing your fifth point, we have restructured the methodology section to provide a holistic view of the proposed work. The method is now described using a hierarchical, component-based design, clearly explaining the interactions and concurrency between the components. Furthermore, we have added a explanation how our approach achieves better computational efficiency through dynamic parameter adaptation and an enhanced hill climbing algorithm.

6) Finally, to address your sixth point, we have significantly improved the results section. It now includes subsections covering the organization of overall experiments (subsection 5.1), experimental setup (subsection 5.3), benchmark instances and parameters (subsection 5.2), performance metrics (subsection 5.4), visualization and explanation of results (subsection 5.5), comparison with state-of-the-art techniques (subsection 6.1), significance of achieved results (subsection 6.2), and limitations of the results (subsection 6.3). These improvements provide a comprehensive and structured presentation of our experimental findings.

We believe that the revised manuscript, incorporating the changes based on your valuable feedback, has substantially enhanced the quality, clarity, and impact of our work. Thank you once again for your time and expertise in reviewing our manuscript.

Best regards,

Oleksandr Kuznetsov

Reviewer 3 Report

Comments and Suggestions for Authors

After the first round of reviewing, the quality of the manuscript was greatly improved, making it almost ready for publication in its current form: only a few minor issues should be addressed, but they only concern punctuation.

The authors have efficiently implemented the changes suggested/recommended in the first review round, addressing the main criticisms present in the original version of the manuscript, in particular the implications on fundamental security aspects due to the correlation between the non-linearity of the studied S-boxes (substitution boxes) and the power analysis attacks (which are a category of side-channel attacks). Also, the addition of Table 1 (which was not present in the original version), which reports a comparison between the existing heuristic techniques for S-box generation, has greatly improved the readability of the manuscript content. Similarly, the readability was also improved by addressing the minor points regarding the quality of English and spelling, which greatly improved the readability of the paper.

In conclusion, the content of the manuscript in its current form is excellent. Only a few additional changes can be made to further improve the quality of the format, as listed below.

*) Line 604. I think a semicolon is missing to separate the number -28 and the number -24: “X=-32;-28-24;-20;-16;-12;-8;-4;0;4;8;12;16;20;24;28;32;36;40;44” maybe should be “X=-32;-28;-24;-20;-16;-12;-8;-4;0;4;8;12;16;20;24;28;32;36;40;44”.

*) All figures except figure 1 include a final period in the corresponding caption, while all tables except table 6 do not include a final period in the corresponding caption. It is suggested to unify the format of the caption for both figures and tables by deciding whether or not to include the final period in the caption for all figures and all tables. Both solutions (include or not) are valid. As a preference, I would suggest including the final period in the caption.

Author Response

Dear Reviewer,

We are grateful for your thorough review and insightful comments on our revised manuscript. We also appreciate your acknowledgment of our efforts to address the minor points related to the quality of English and spelling.

Regarding the formatting issues you mentioned, we have made the following changes:

  • In line 604, we have added the missing semicolon to separate the numbers -28 and -24 in the sequence.
  • We have unified the format of the captions for both figures and tables by including a final period in all captions, as per your suggestion.

Thank you once again for your valuable feedback and recommendations. Your input has been crucial in refining our manuscript.

Best regards,

Oleksandr Kuznetsov

Reviewer 4 Report

Comments and Suggestions for Authors

Accepted in the present form

Author Response

Dear Reviewer,

We are delighted to learn that our manuscript is accepted in its present form. We would like to express our sincere gratitude for your time and effort in reviewing our work.

Thank you for your support and for recommending our work for publication.

Best regards,

Oleksandr Kuznetsov

Round 3

Reviewer 2 Report

Comments and Suggestions for Authors

Authors have addressed the raised concerns. 

Article can be published in its current form